# Global site-specific N-glycosylation analysis of HIV envelope glycoprotein

Liwei Cao[1], Jolene K. Diedrich[2], Daniel W. Kulp[3,4], Matthias Pauthner[3,4], Lin He[2], Sung-Kyu Robin Park[2], Devin Sok[3,4], Ching Yao Su[3], Claire M. Delahunty[2], Sergey Menis[4], Raiees Andrabi[3], Javier Guenaga[3], Erik Georgeson[3], Michael Kubitz[3], Yumiko Adachi[4], Dennis R. Burton[3,4], William R. Schief[3,4], John R. Yates III[2] & James C. Paulson[1,2,3]

HIV-1 envelope glycoprotein (Env) is the sole target for broadly neutralizing antibodies (bnAbs) and the focus for design of an antibody-based HIV vaccine. The Env trimer is covered by ~90 N-linked glycans, which shield the underlying protein from immune surveillance. bNAbs to HIV develop during infection, with many showing dependence on glycans for binding to Env. The ability to routinely assess the glycan type at each glycosylation site may facilitate design of improved vaccine candidates. Here we present a general mass spectrometry-based proteomics strategy that uses specific endoglycosidases to introduce mass signatures that distinguish peptide glycosites that are unoccupied or occupied by high-mannose/hybrid or complex-type glycans. The method yields >95% sequence coverage for Env, provides semi-quantitative analysis of the glycosylation status at each glycosite. We find that most glycosites in recombinant Env trimers are fully occupied by glycans, varying in the proportion of high-mannose/hybrid and complex-type glycans.

[1] Department of Cell and Molecular Biology, The Scripps Research Institute, La Jolla, California 92037, USA. [2] Department of Chemical Physiology, The Scripps Research Institute, La Jolla, California 92037, USA. [3] Department of Immunology and Microbial Science, The Scripps Research Institute, La Jolla, California 92037, USA. [4] Department of the IAVI Neutralizing Antibody Center, The Scripps Research Institute, La Jolla, California 92037, USA. Correspondence and requests for materials should be addressed to J.C.P. (email: jpaulson@scripps.edu).

Despite intensive efforts, no effective human immunodeficiency virus type 1 (HIV-1) vaccine has yet been developed. However, there is promise for the future since broadly neutralizing antibodies (bNAbs) have been shown to protect against virus challenge in animal models[1–3]. Because HIV-1 envelope glycoprotein (Env) is the target for bNAbs, it is an attractive immunogen for eliciting a bNAb response[1,4–6]. Each protomer of the HIV-1 Env trimer contains 26–30 N-glycans that shield most of the polypeptide surface from immune surveillance[7–10]. More importantly, many bNAbs show interactions with specific N-linked glycans on Env that are required for binding and neutralization potency[11–18]. For example, structures of glycan-dependent bNAbs of the PGT121 family in complex with Env have revealed a supersite comprised of a dense array of glycans that are predominately high-mannose-type[11,15–17]. In contrast, bNAbs PG9/PG16 recognize an epitope on the V1V2 loop involving a $Man_5GlcNAc_2$ glycan at N160 and a terminal sialylated complex-type glycan at N156 or N173 (ref. 18). Neutralization studies using glycan deletion mutants of HIV-1 pseudoviruses demonstrated the importance of glycans at specific glycosylation sites for the binding and neutralization of other bNAbs[12,19]. Adding to the complexity, the glycan structures on Env have been shown to be influenced by various factors, such as the type of cell lines used to generate Env[20–22], purification methods[23], structural constraints[23,24], and whether Env is purified from virus, pseudovirus, or expressed recombinantly[25,26]. For all these reasons a robust and routine method for analysis of glycosylation status at each glycosite of HIV-1 Env to support rational design and development of vaccine immunogens is needed.

Analysis of the site-specific glycosylation of HIV Env is challenging due to the fact that the Env gp160 monomer contains 26–30 N-linked glycosylation sites with multiple glycan structures at each site[24,27–31] and millions of different Env sequences in the viruses that are circulating worldwide[17]. Over the past decade mass spectrometry-based methods have yielded significant insights[24,25,27–33]. HIV Env contains a mixture of both high-mannose-type glycans ($Man_{5-9}GlcNAc_2$-Asn), and glycans that are further processed to the complex-type with a $Man_3GlcNAc_2$-Asn core, 2–4 branches and optionally terminated with sialic acid. Notably, some glycosylation sites are exclusively high-mannose while others are almost exclusively complex-type, demonstrating that the host cell processing machinery differentially processes the glycans at each site[24,27–31]. Comparison of recombinant membrane gp160 Env trimer with the cleaved Env (gp120/gp41) analogous to the form in infectious virus revealed that gp160 glycans undergo a much higher degree of processing to complex-type glycans, which corresponded to more open configurations of the gp160 Envs that could provide greater access for the glycan processing machinery[23]. Immunization studies further showed that cleaved Env, but not uncleaved Env, immunogens induced neutralizing antibodies (NAbs) against the autologous tier 2 viruses[5].

In the most thorough study of Env glycosylation to date, Crispin et al.[22] employed matrix-assisted laser desorption/ionization–time of flight mass spectrometry (MS) and liquid chromatography-tandem mass spectrometry (LC-MS/MS) to analyse site-specific glycosylation of BG505 SOSIP.664 trimer, resulting in the identification of 20–26 out of 28 glycosites[24,34]. The method involved separately elucidating the many glycans present in the mixture cleaved from the intact Env trimer. Glycosylated peptides were purified using hydrophilic interaction chromatography-based enrichment and analysed by LC-MS/MS searching for molecular weight species corresponding to molecular weigths of glycopeptides with the library of glycans, providing a detailed analysis of glycoforms for the sites detected.

Because non-glycosylated peptides were discarded, information about the site occupancy was absent in this study. While the method requires time consuming manual interpretation of a large volume of data collected during LC-MS/MS, it provides high quality detailed information about glycans at each of the glycosites detected[24].

Here we present a method that enables us to assess the site occupancy and proportion of high-mannose and complex-type glycans at all glycosites of HIV-1 Env, and is in principle applicable to any glycoprotein. The method employs digestion with multiple proteases to improve sequence coverage and generate multiple peptides for each glycosylation site. Peptide mixtures are treated sequentially by two endoglycosidases, Endo H followed by PNGase F, resulting in the introduction of novel masses for peptides that contain high-mannose glycans, complex-type glycans or no glycan. Of note, if hybrid type glycans are present in target glycoproteins, they are included in the category of high-mannose glycans as Endo H is able to release both of high-mannose and complex-type glycans. Data is analysed by powerful proteomics software that has been optimized to identify and calculate the abundance of each peptide using ion intensity peak area of MS signals. This reduces the manual analysis of data and provides a semi-quantitative assessment of the three states of glycosylation for each glycosite. Here we evaluate the impact of various factors on the glycosylation of HIV-1 Env, including purification methods, quaternary structure, glycosite mutations, and alternative patterns of glycosites generated by different virus strains. The results show that this method serves as a robust tool for characterization of site-specific glycosylation of HIV-1 Env to facilitate the rational design and development of vaccine immunogens.

## Results

**Strategy**. We devised a proteomics-based strategy to assess the degree of glycan processing and the degree of site occupancy of each glycosite, defined as the N-glycosylation sequon N-X-S/T (Fig. 1a). To this end, we used sequential treatment with two endoglycosidases common to glycoproteomics workflows to introduce novel mass signatures for peptides that contain the two types of glycans[29,35,36]. First, Env peptides are digested with Endo H to cleave high-mannose (and hybrid) glycans between the innermost GlcNAc residues, leaving a GlcNAc attached to the Asn (N + 203). Subsequent PNGase F treatment removes any remaining complex-type glycans, and in the process converts Asn to Asp resulting in a + 3 Da mass shift when the reaction is conducted in $O^{18}$-water (N + 3)[37–41]. For peptides with unoccupied glycosites these treatments produce no mass shift (N + 0). With this strategy, the analytical problem is focused on generating suitable peptides and assessing the relative abundance of the three states of glycosylation.

To maximize sequence coverage, we employed digestion with multiple proteases. Each sample was subjected to chymotrypsin, trypsin and chymotrypsin, or 'triple digestion' developed by Gatlin et al.[42] involving digestion with trypsin and the nonspecific proteases, elastase and subtilisin (Fig. 1b). Each of the three proteolytic digestions was conducted in triplicate. Proteases were denatured by heating to prevent incorporation $O^{18}$ at the C-terminus in subsequent endoglycosidase treatments[37]. Peptides were digested sequentially with Endo H and PNGase F in $O^{18}$-$H_2O$ to generate novel masses for the three states of glycosylation as described above. Conditions for each enzyme digestion were optimized to ensure complete digestion.

For each of the nine samples, LC-MS/MS data were acquired. Peptides were identified using ProLuCID[43–46]. The abundance of each peptide, as determined by the sum of the peak areas from all

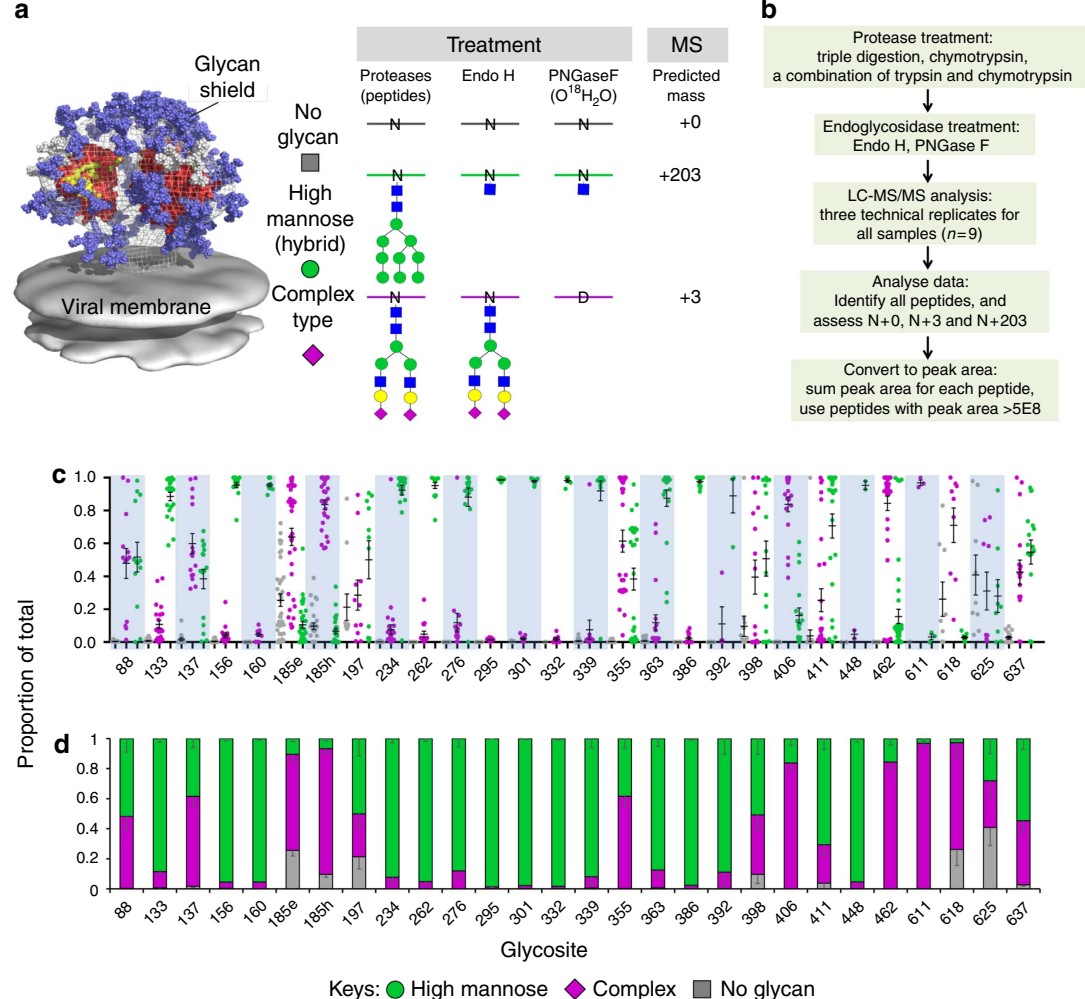

**Figure 1 | Schematic representation of a method and to determine site-specific glycosylation of BG505 SOSIP.664 trimer.** (**a**) Introduction of novel masses for peptides with glycosites that contain high-mannose (hybrid) glycans, complex-type glycans, or are not glycosylated, by using Endo H treatment followed by PNGase F deglycosylation. (**b**) The workflow of the method present in this study. (**c**) Scatter plot of the site-specific glycosylation of BG505 SOSIP.664 trimer purified with $Ni^{2+}$/SEC. A set of peptides with N + 0 (grey dots), N + 3 (purple dots), and N + 203 (green dots) modifications that were identified when at least one of the three had a peak area of at least > 5E8 was displayed. (**d**) Colour-coded bar graph of the site-specific glycosylation of BG505 SOSIP.664 trimer. Mean ± s.e.m. were plotted.

identified charge states[24], was calculated using Census[47]. Ion injection time was used to further normalize the resulting peak area. In order to improve the accuracy of the method, a set of peptides with N + 0, N + 3, and N + 203 modifications were considered only when at least one of the three had a peak area of at least 5E8. This value was determined empirically as optimal to distinguish information from spectral noise as discussed further in Supplementary Fig. 5. For each glycosite site, the proportion of high-mannose, complex-type and unoccupied was determined for each peptide containing that glycosite detected in each MS run. Then the results were averaged over all peptides in all MS runs of that sample, with the mean and standard error used as the final result for proportion of the three glycosylation states at each site. In the event that no peptide for a given glycosite met the threshold of 5E8, spectral hits were manually assessed as a basis for quantitation for that glycosite only. Otherwise, no manual interpretation of the data was needed, allowing for high-throughput analysis of the glycosylation of HIV-1 Env.

To test the general utility of the method, we analysed three model glycoproteins, bovine fetuin, invertase produced by the yeast *Saccharomyces cerevisiae*, and a recombinant influenza virus haemagglutinin (HA) from H3N2 strain A/Victoria/361/2011 produced in HEK 293 F cells[48]. As expected, the three N-linked glycosites (Asn-X-Thr/Ser) of fetuin were entirely complex-type glycosylation, and site occupancy was > 99% for two sites (N99, N156) and 89% for the third site (N176; Supplementary Fig. 1), consistent with the previous report on the glycosylation of bovine fetuin[49]. In contrast, the 14 N-glycosites of invertase produced by the yeast *S. cerevisiae* were completely high-mannose type glycosylation, and site occupancy was only 42 and 77% for the sites N275 and N64, respectively, while the remaining 12 were with > 90% site occupancy (Supplementary Fig. 2). Of the 12 glycosites in the influenza haemagglutinin of A/Victoria/361/2011, one was not occupied (N122), another was only 32% occupied (N144) and the remaining ten were fully occupied, in line with glycosylation site occupancy of H3N2 strain observed in previous studies[50,51]. Of those that were glycosylated, four were completely complex-type (N22, N38, N63 and N483), two were > 95% high-mannose (N45 and N285) and the rest were a combination of high-mannose and complex-type (N126, N133, N165 and N246) (Supplementary Fig. 3). For glycosites N165 and N246 that are associated with interactions

between HA and SP-D lung collectin, we found that both of them were mainly occupied by high-mannose glycans, in agreement with the observations in previous studies[50,51]. Although peak areas are primarily used for quantitation, there were 85–699 spectral hits detected for each glycosylation site as summarized in Supplementary Tables 1 and 2.

We next applied this method to assess the site-specific glycosylation of the soluble recombinant HIV-1 Env trimer BG505 SOSIP.664 (ref. 52), the structure of which has been determined by crystallography and cryo-EM[11,53–55]. To avoid biasing the specific glycan structures during purification, we started with C-terminal His-tagged trimer purified by $Ni^{2+}$-NTA affinity chromatography followed by size exclusive chromatography ($Ni^{2+}$/SEC). Each of the 28 glycosites on the Env monomer was identified with multiple MS/MS spectra and spectral hits ranging from 90 to more than 2,000 per site (Supplementary Table 3). Because both specific and nonspecific proteases were used, multiple peptides were identified for each glycosite. Each peptide generates a set of up to three endoglycosidase-processed peptides representing the three states of glycosylation: unoccupied (N + 0), high-mannose (N + 203) and complex-type (N + 3). Previous analysis of N-glycans of recombinant BG505 SOSIP.664 trimer revealed small amounts of hybrid glycans (3–20%) at several glycosites[24], and in our analysis these would be included in the high-mannose category (N + 203) due to the specificity of Endo H. The proportion of the three states of glycosylation for peptide sets that reached the threshold peak area of 5E8 are shown in a scatter plot in Fig. 1c. The results suggest that the sum of the information for all peptides at a single glycosite will provide a much more unbiased assessment of the glycosylation status than any single peptide. For example, with sites that have a mixture of both high-mannose and complex-type glycans (for example, 137, 355), most peptide sets show a mixture of the two types, but a few show completely high-mannose or complex-type. Even more extreme, while most of the peptide sets at glycosites N234, N276 and N339 detect predominantly high-mannose glycans (+ 203), one peptide set for each site detected only complex-type glycans (+ 3). These observed biases did not appear to be due the specific proteases used (Supplementary Fig. 4). Such analysis underscores the importance of analysing multiple sets of peptides for each glycosite to avoid erroneous conclusions. Accordingly, as an unbiased measure of all the data, we have used an average value calculated from all peptide sets for each glycosite. The results are displayed in a colour-coded bar graph that allows visual comparisons between glycosites (Fig. 1d).

The results provide a robust semi-quantitative analysis of site-specific glycosylation of HIV-1 Env at each glycosylation site. It is immediately evident that the majority of the BG505 SOSIP glycosites are fully glycosylated. All but four out of the 28 sites are greater than 90% occupied and none are >50% unoccupied. Previous studies on site-specific analysis have not included an analysis of site occupancy[24], or have conducted qualitative analysis that imply site occupancy to be much less[28]. Although most sites have a mixture of high-mannose and complex-type glycans, 14 sites have >75% high-mannose, six sites have >75% complex-type, and the remaining eight sites have a mixture of both types. In summary, it is clear that processing of glycans from high-mannose type to complex-type by the host cell glycosylation machinery is highly site-specific.

The results for all 28 glycosites are summarized using a model of the BG505 SOSIP trimer with glycans colour-coded using the system proposed by Behrens et al.[24] to distinguish glycosites with predominately high-mannose-type, complex-type, or a mixture of the two (Fig. 2). For the glycosites detected and analysed by Behrens et al.[24,34] we find excellent qualitative agreement in the

proportion of high-mannose and complex glycans found in the BG505 SOSIP.664. The high-mannose patch comprised of glycosites N295, N332, N339, N386 and N392 and the adjacent glycans at N301 and 411 are almost exclusively high-mannose structures[17,56]. Previous structural studies have suggested that a sialylated glycan present at site N156 plays a key role for PG9/PG16 binding[13,57], so it was surprising that this site was mainly occupied by high-mannose structures. One potential explanation is that glycans on V1/V2 scaffolds crystallized with PG9 and PG16 may be processed differently than those on native Env trimer[23,24]. On the basis of PGT121 recognition of complex-type glycans in the glycan array, the crystal structures of PGT121 bound to complex glycans, and the proximity of PGT121 to glycosite N137, it was predicted that N137 would be complex-type[11,15,58]. Here we showed that this site is indeed mainly occupied by complex-type glycans (~60%). Glycosite N276 on the edge of the CD4-binding site[24,56], and the highly conserved glycan N262 needed for folding of gp120 (ref. 59), were both found to be exclusively high-mannose glycans. The epitope of bNAb PGT151 is comprised of the gp120-gp41 interface of one protomer and glycans N611 and N637 of the adjacent protomer that were predicted to be complex-type glycans[12,60], consistent with results here. The other glycosites located in the gp41 region, N618 and N625, were found to be partially non-glycosylated, in keeping with earlier biochemical studies suggesting incomplete of occupancy glycosites in gp41 (refs 61,62).

**Validation of MS detection of glycotypes.** The current method employs the specificity of endoglycosidases to introduce novel masses for peptides that contain high-mannose or complex-type glycans. One major assumption is that enzyme treatments would go to completion, avoiding miss-assignment of the glycosylation status. For example, if high-mannose glycans were not completely removed by Endo H and were subsequently cleaved by PNGase F, they would be scored as complex-type glycans. Thus, we first optimized digestion conditions to completely remove glycans on whole glycoproteins (for example, fetuin and invertase) as monitored by gel shift assays. Another major assumption is that the endoglycosidase-processed peptides with glycosites (Asn-X-Thr/Ser) representing the three glycosylation states with Asn (N + 0; unoccupied), GlcNAc-Asn (N + 203; high-mannose) or Asp (N + 3) would be detected equally during LC-MS/MS analysis[63]. These assumptions were directly assessed with a strategy outlined in Fig. 3a using BG505 SOSIP.664 trimer expressed in the presence of kifunensine (Kif_BG505), which should be exclusively occupied by high-mannose glycans[25]. High-mannose glycans on Kif_BG505 were either removed by Endo H followed by PNGase F, generating homogeneous N + 203, or alternatively by PNGase F treatment only, generating N + 3 tags. With Endo-H only, glycosites comprised >95% N + 203 (green bars), confirming that the sample prepared in the presence of Kif had high-mannose glycans, and that Endo H treatment was highly effective (Fig. 3b). The low percentage of N + 3 (purple bars) could be either from incomplete removal of high-mannose glycans by Endo H, incomplete inhibition of glycan processing by Kifunensine, or false positive results generated from data processing. With PNGase F treatment only, mimicking a sample with entirely complex-type glycans, >98% of the glycosites were N + 3 (Fig. 3c). We presume low percentages of N + 203 appearing at several glycosites, (for example, N398 and N137) are likely false positives generated from data processing. The percentages of false positives increased when the threshold of peak area was reduced from 5E8 to 5E7 (Supplementary Fig. 5), whereas further raising this threshold to 5E9 resulted in losing signals from some glycosites.

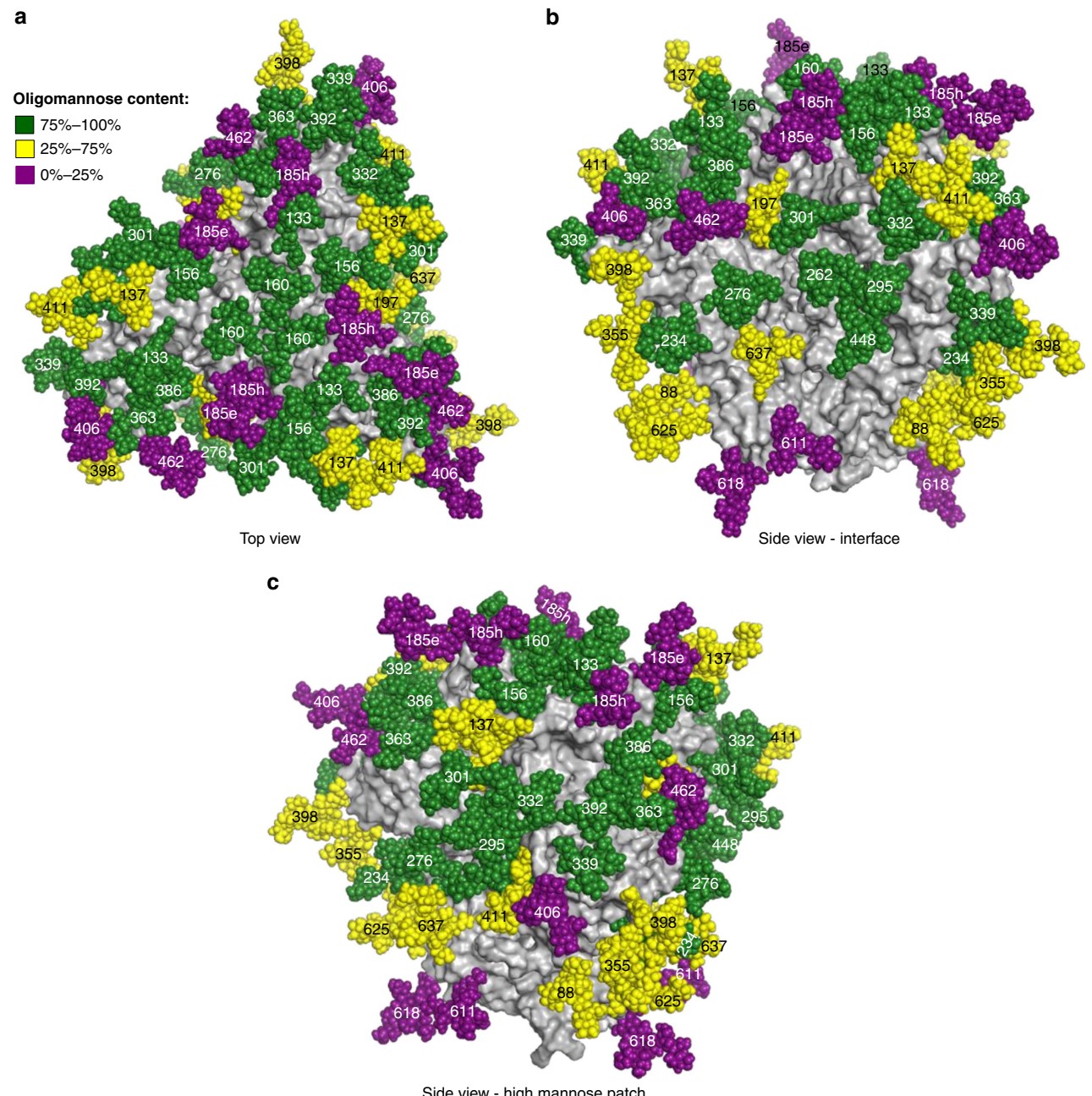

**Figure 2 | Mapping of site-specific glycosylation onto the structure of BG505 SOSIP.664 trimer.** The full glycosylated model was constructed using PDB: 4TVP, RosettaRemodel and GlycanRelax[88–90]. The trimer is shown as a grey surface and the glycans are shown as spheres coloured by proportion of oligomannose content at that site. (**a**) Top view of the model. (**b**) Side view of the model: the interface of two protomers. (**c**) Side view of the model: the high-mannose patch.

Having two samples containing Env peptides that are identical except for glycosites with N + 203 (Fig. 3b) or N + 3 (Fig. 3c), we could assess MS detection efficiency by mixing them at a molar ratio of 1:1 prior to LC-MS/MS analysis (Fig. 3d). The results show that both the N + 203 and N + 3 glycosites were detected for each site, with a ratio of 1.0 to 1.2, respectively, suggesting slightly increased sensitivity for the N + 3. The red dotted line represents the average proportion of high-mannose and complex-type glycans obtained in this experiment. For the most part, the site-specific ratios of high-mannose and complex glycans were similar to the average for most glycosites. To determine how corrections for differences in detection would impact the results obtained for the BG505 SOSIP.664 trimer shown in Fig. 1d, we

first applied site-specific corrections for high-mannose and complex-type glycans (Supplementary Fig. 6b). The corrected values were within the standard error of uncorrected values except for a single site, N637, and even for that site, the proportion of high-mannose and complex-type glycans were qualitatively similar before and after correction (Supplementary Fig. 6a,b). Because spectra counts are often used as a quantitative measure, we assessed spectral counts for the three forms of the peptides ( + 0, + 3, + 203) to determine the relative abundance of no glycan, high-mannose and complex-type at each glycosite (Supplementary Fig. 6c). In contrast to the site-specific correction, quantitative assessment using spectral counts gave significant differences at 14 out of 28 sites. In every case, there

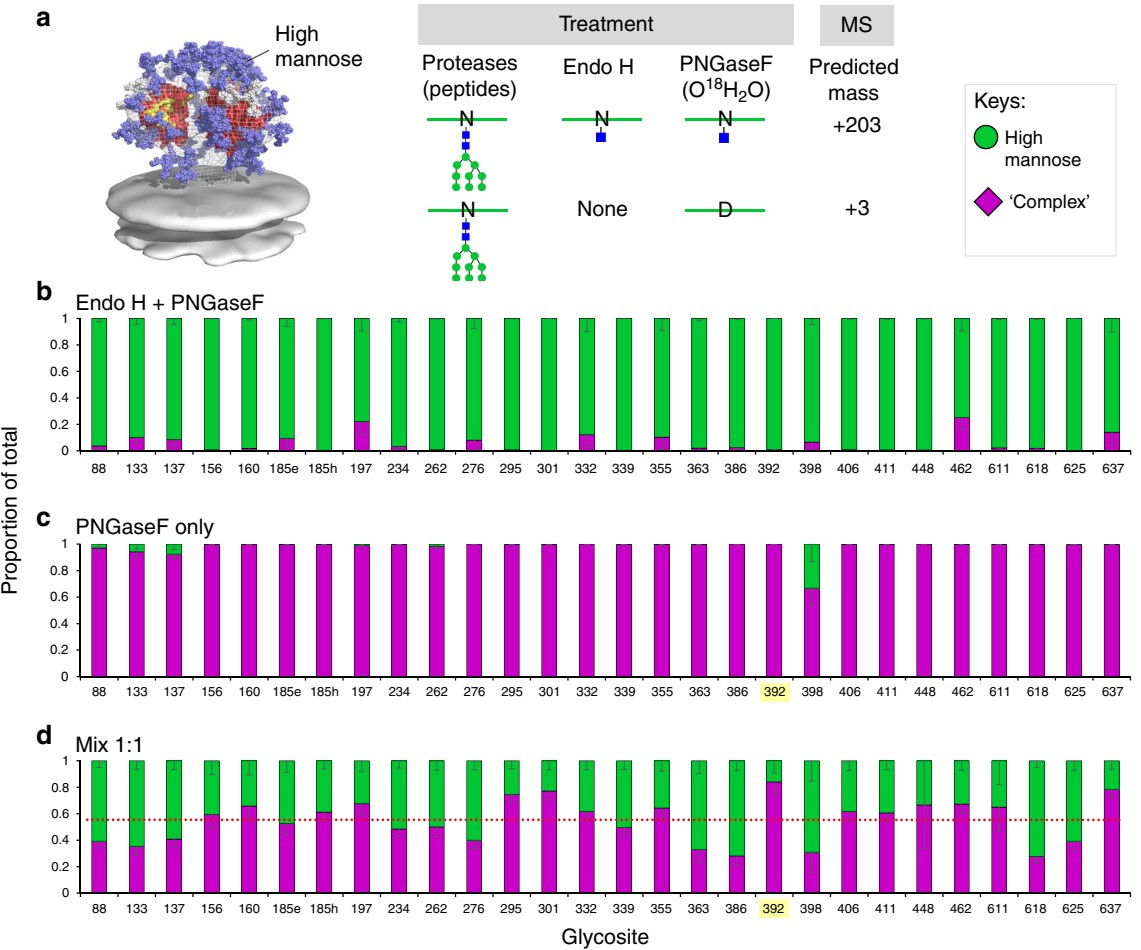

**Figure 3 | Validation of endoglycosidase treatments and MS detection for peptides that contain N + 3 and N + 203 modifications.** (**a**) Schematic representation of the process. (**b**) Validation of Endo H treatment by using Kif_BG505. (**c**) Validation of PNGase F treatment was conducted by using Kif_BG505 without Endo H treatment. (**d**) MS detection for peptides that contain homogeneous N + 3 and N + 203 modifications at a molar ratio of 1:1. Peptides that had potential glycosites, but were not glycosylated were not included. The proportions of high-mannose and complex-type glycans at those glycosites highlighted in yellow were assigned based on the proportion of spectra hits since peak area did not reach the threshold of 5E8 Supplementary Tables 5–7.

was an apparent increase in the proportion of complex-type glycans particularly in glycosites that are known to be occupied predominantly by high-mannose glycans[24,56], such as N156, N262, N295, N332, N339, N386 and N392. We attribute to sensitive detection, since counts for low and high abundant species would be counted as equivalent, giving a bias for the lower abundant complex-type glycans. For these reasons, we have used peak area for all results presented, and have not applied site-specific corrections since they were not found to significantly change the result (Supplementary Fig. 6b).

**The impact of purification methods on Env glycosylation.** With the method established we investigated the influence of several factors that could affect results from recombinant HIV-1 Env samples. We first assessed the potential for batch variation in preparations of the BG505 SOSIP.664 trimer with a C-terminal hexa histidine-tag produced in 293 F HEK cells purified using $Ni^{2+}$-NTA affinity chromatography followed by size exclusion chromatography. Compared with the reference batch (Fig. 1d) we saw virtually identical results with the second batch (Fig. 4a), with only a single glycosite (N137) showing significantly, but only marginally, lower proportion of complex-type glycans.

Use of glycan-dependent bNAbs for purification of HIV-1 Env raises the prospect of selection of and enrichment of particles that carry a favourable glycan recognized by the antibody[22,23]. To test this possibility directly, we compared the glycosylation of the $Ni^{2+}$-NTA affinity column-purified material (Fig. 1d) with the same batch of recombinant BG505 SOSIP.664 trimer that was alternatively purified using affinity columns using bNAbs 2G12 (ref. 64), PGT145 (ref. 65) or PGT151 (refs 12,60), in each case followed by size exclusion chromatography (Fig. 4b). We found that BG505 SOSIP.664 trimer purified using 2G12, PGT145 or PGT151 had an almost identical glycosylation pattern compared with the trimer purified using $Ni^{2+}$-NTA affinity chromatography, indicating that different purification methods have little impact on the glycosylation pattern of BG505 SOSIP.664 trimer.

**Impact of glycans on compositions at neighbouring glycosites.** Mutants of HIV-1 pseudovirus, with specific glycan sites removed, have been widely used in neutralization studies to assess the degree to which specific glycans are involved in glycan-dependent bNAbs epitopes[12,15,19]. Furthermore, design of candidate vaccine priming immunogens to activate bNAb

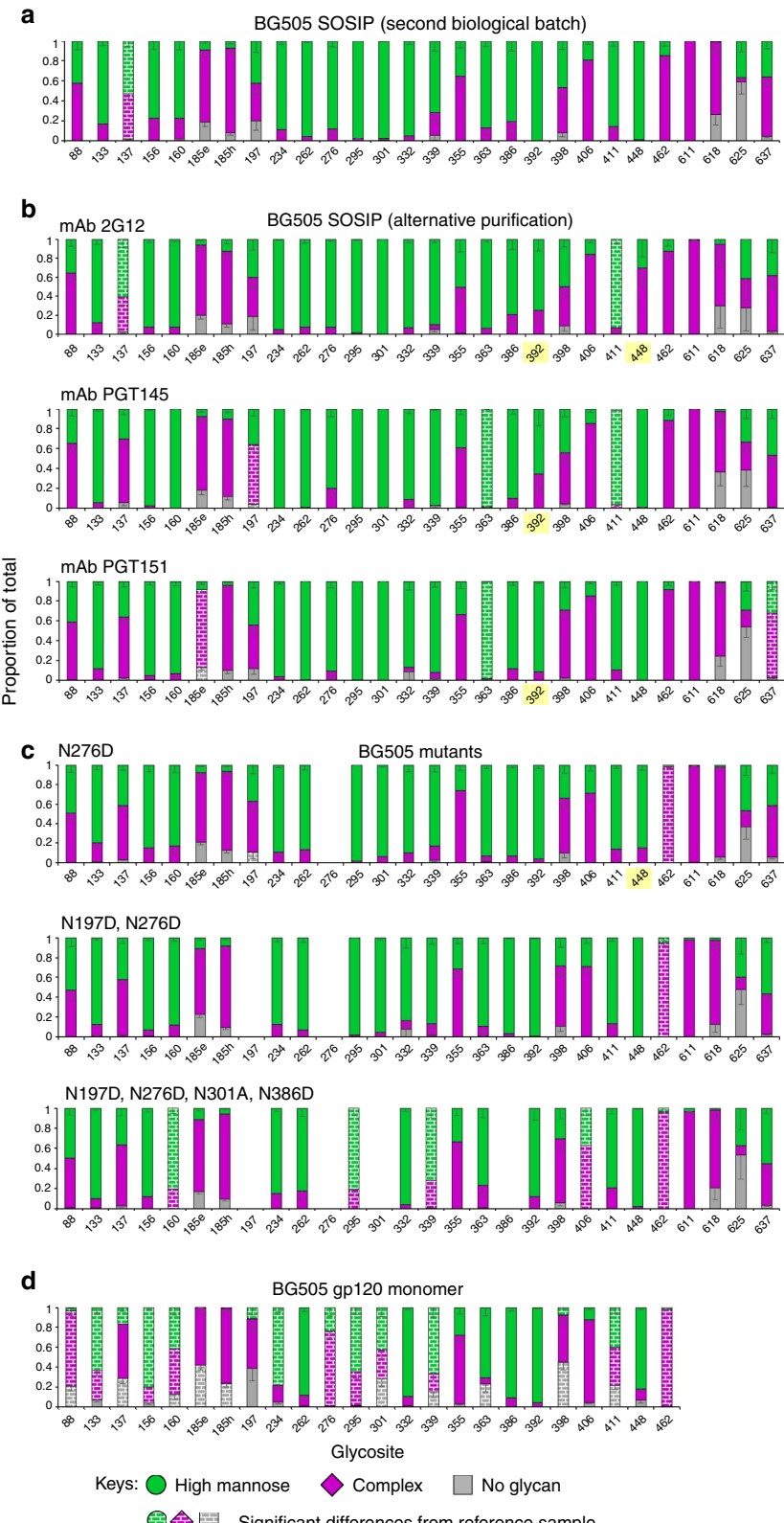

**Figure 4 | Site-specific glycosylation of BG505 Envs.** (**a**) second biological batch of BG505 SOSIP.664 trimer that was purified with $Ni^{2+}$/SEC; (**b**) BG505 SOSIP.664 trimers that were purified over different bNAbs columns followed by SEC; (**c**) the mutants of BG505 SOSIP.664 trimer; (**d**) BG505 gp120 monomer. The glycosylation of BG505 SOSIP.664 trimer shown in Fig. 1d was selected as reference for comparison. The proportions of high-mannose and complex-type glycans at those glycosites highlighted in yellow were assigned based on the proportion of spectra hits since peak area did not reach the threshold of 5E8 (Supplementary Tables 8–23).

precursor B cells has generally involved removal of one or more glycans from Env or Env-derived proteins[66–71]. A gap in current knowledge is the impact of removing glycosylation sites on neighbouring glycosites and the overall glycosylation of Env. To this end, we analysed three mutants of recombinant BG505 SOSIP.664 trimer with the deletion of 1–4 glycan sites near the CD4bs, including glycan sites N276, N197, N301 and N386 (refs 21,24,72; Fig. 4c). When compared with the glycosylation of BG505 SOSIP.664 trimer (Figs 1d and 4a), deletion of glycan site at N276 (N276D) produced only slight but significant changes at N462 and N197. Further deletion of site N197 (N197D, N276D) gave a virtually identical glycosylation pattern. The mutant with four glycosites removed (N197D, N276D, N301A and N386D) showed significant differences at five sites (N160, N295, N339, N406 and N462), but again the differences were slight, and qualitatively the same as the fully glycosylated reference BG505 trimer (Figs 1d and 4a). Thus, perhaps surprisingly, removal of glycans close to CD4bs did not change access of the glycan processing machinery sufficiently to alter the ratio of total high-mannose/hybrid and complex-type glycans at neighbouring sites, or the overall glycosylation of the BG505 SOSIP.664 as assessed by this method.

**The impact of quaternary structure on Env glycosylation**. Recombinant monomeric gp120 has been used as a common immunogen in various vaccine trials[22,73]. Structural constrains imposed by trimerization have been shown to impact the glycosylation of HIV-1 Env, resulting in higher content of high-mannose glycans in the trimer[22,25]. To assess the impact of trimerization on site-specific glycosylation, we assessed the glycosylation of the BG505 monomer (Fig. 4d). In contrast to minimal differences in glycosylation of the BG505 trimer glycosylation mutants, significant changes in glycosylation were observed in 17 out of the 24 glycosites of the monomer relative to the trimer. As observed by others, the monomer had increased levels of complex-type glycans. We also observed a striking decrease in site occupancy at 10 glycosites relative to the trimer. Of note, highly conserved glycosites, for example, N156, N262, N295, N332, N386 and N392, were still predominantly high-mannose structures. Glycans with a dramatic shift to complex-type included N88, N160, N276 and N411 (Figs 1d and 4a,d). Overall, the site occupancy of BG505 monomer gp120 was 86%, similar to that reported previously[74]. The results show that the glycosylation pattern of monomeric gp120 is dramatically different from that of the native-like Env trimer, which can be attributed in part to greater accessibility of glycans at the subunit interface by the glycosylation machinery[25,26]. Such changes will impact the binding of glycan-dependent antibodies and should be considered when using gp120 proteins as immunogens.

**Variation in site-specific glycosylation of Env trimers**. The glycosylation of cleaved, native-like Env trimers derived from diverse genotypes, including JR-FL SOSIP.664 (subtype B)[75], B41 SOSIP.664 (subtype B)[65], CRF02_AG_250 SOSIP.664 (subtype AG)[76], 327c SOSIP.664 (subtype C)[77] and simian immunodeficiency virus (SIV) trimer SIVcpzMT145 SOSIP.664[78] were analysed and compared with that of BG505 SOSIP.664 trimer (subtype A) (Fig. 5). To facilitate comparison of glycosites, sequences were aligned using a reference HIV Env (Supplementary Fig. 7). As a group, the Env trimers have similar glycosylation patterns. Almost all glycosites are fully occupied in the gp120 portion of the six trimers. The conserved glycosites[17], N156, N262, N276, N301, N295, N332/334, N411/413 and N448, were mainly occupied by high-mannose glycans, and in contrast, complex-type

glycans were predominately found at sites N406 and N460-463 and in most of the glycosites of the gp41 subunit.

In addition to the similarities there were also significant differences among the trimers. The proportions of high-mannose glycans ranged from a low of ∼50% for JR-FL SOSIP.664 and CRF02_AG_250 SOSIP.664 to somewhat higher (60–66%) for other isolate trimers. For example, glycans at N137 consisted predominantly of complex-type glycans on most of the trimers, but were largely high-mannose glycans on B41 SOSIP.664 and 327c SOSIP.664. Although complex-type glycans at the glycosite N137 have been shown to be involved in the formation of bNAb epitopes such as those for the PGT121 family[11,15,58], complex-type glycans are apparently not essential at this position for neutralization of these strains. There was also variation in glycosites located in the canonical high-mannose patch, including N339, N386 and N392. For instance, BG505 SOSIP.664 trimer, B41 SOSIP.664, CRF02_AG_250 SOSIP.664 and SIVcpzMT145 SOSIP.664 each were occupied predominantly by high-mannose glycans at the site N386, while a high proportion of complex-type glycans was found at this site on two other trimers. In addition, high-mannose glycans dominated the site N356 on SIVcpzMT145 SOSIP.664, whereas a high proportion of complex-type glycans was detected at this site on five other trimers. There were also variations in the site occupancies and types of glycans in the gp41 region among the six trimers. For instance, the glycosite N637 was mainly occupied by high-mannose glycans on BG505 SOSIP.664 trimer, JR-FL SOSIP.664, B41 SOSIP.664 and 327c SOSIP.664, while a high proportion of complex-type glycans was found at this site on CRF02_AG_250 SOSIP.664 and SIVcpzMT145 SOSIP.664. The glycosites located on gp41 subunits were almost completely occupied on SIVcpzMT145 SOSIP.664, while this region was largely unoccupied on other trimers. In summary, the results show a conservation of an overall glycosylation pattern for Envs of different HIV strains, but there are also significant differences.

## Discussion

We have developed a mass spectrometry-based workflow for global analysis of protein glycosylation that determines the degree of glycan occupancy and the proportion of high-mannose/hybrid and complex-type glycans at each glycosite. Although used here to analyse HIV Env, the method is in principle applicable to any glycoprotein. The strategy employs a combination of proteases, including triple digestion, chymotrypsin and a combination of trypsin and chymotrypsin, to generate multiple overlapping peptides, resulting in >95% sequence coverage of HIV-1 Env. For those glycoproteins with fewer N-glycosites, such as fetuin and invertase, triple digestion alone is able to generate detectable peptides that contain all N-glycosites for MS analysis. A combination of triple digestion and chymotrypsin would be enough for detecting all N-glycosites in most glycoproteins. Sequential treatments with Endo H and PNGase F were employed to create different mass signatures at the Asn of each glycosite based on occupancy with high-mannose/hybrid glycan (N + 203), complex-type glycan (N + 3) or no glycan (N + 0). We could then apply standard database searching and label-free analysis[43–45,47] to conduct high-throughput identification and semi-quantitative analysis of peptides representing the three glycosylation states of each glycosite. A complete analysis can be accomplished in a few days with only 30 μg of glycoprotein, representing a small fraction of the material and time needed to do a more detailed glycomics analysis to identify all glycoforms at each site.

Several groups have conducted detailed analysis of Env glycosylation by LC-MS/MS analysis of intact glycopeptides[24,27–31,79]. In contrast to the method described here, glycoproteomics

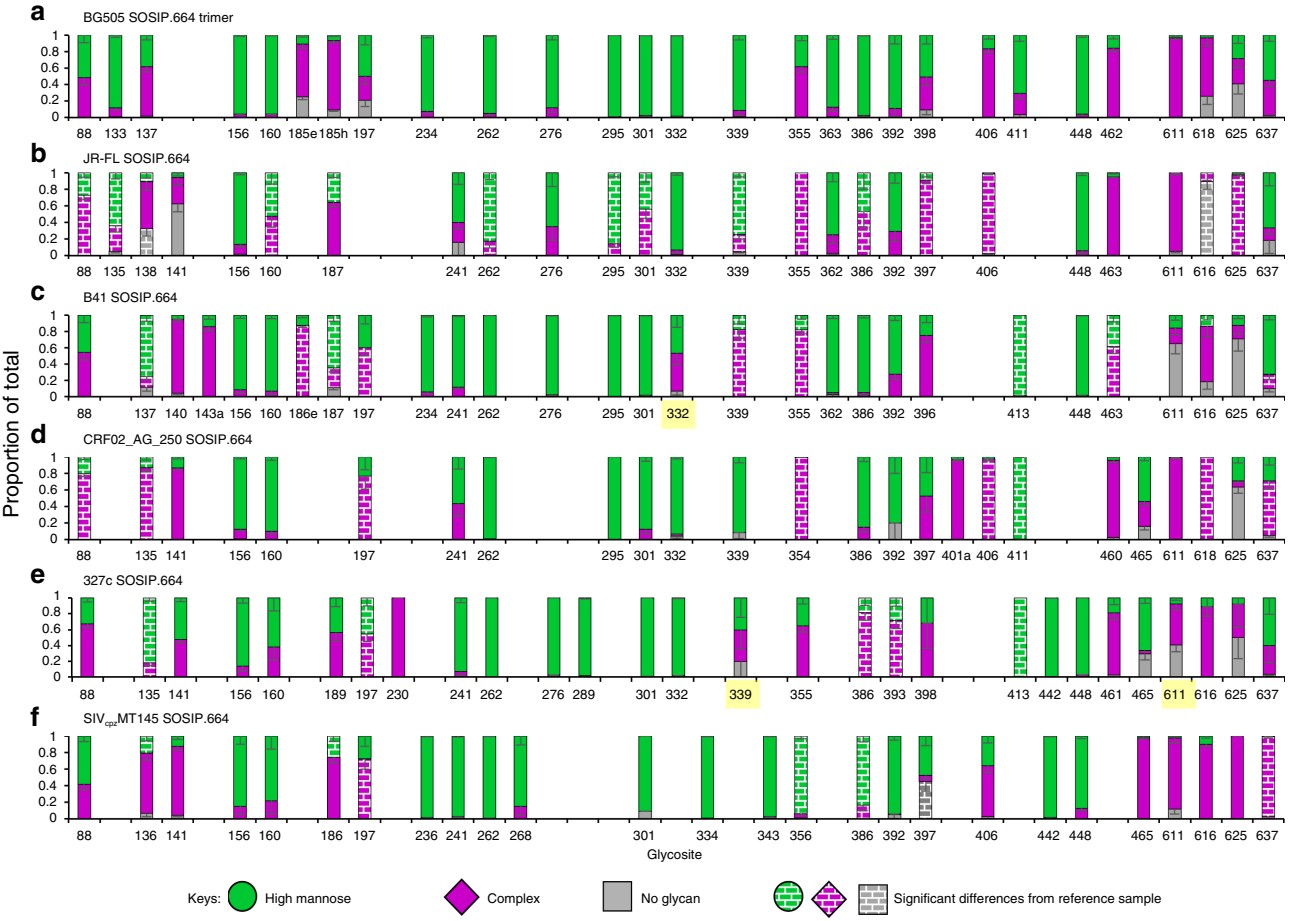

**Figure 5 | Site-specific glycosylation of native-like Env trimers from different HIV-1/SIV isolates.** (**a**) BG505 SOSIP.664 trimer (subtype A), which is the same data as shown in Fig. 1d; (**b**) JR-FL SOSIP.664, subtype B; (**c**) B41 SOSIP.664, subtype B; (**d**) CRF02_AG_250 SOSIP.664, subtype AG; (**e**) 327c SOSIP.664, subtype C; (**f**) SIV trimer SIVcpzMT145 SOSIP.664. N-linked glycosites from multiple HIV Env trimers were aligned according to their relative position to HXB2 and their predicted functional importance. The glycosylation of BG505 SOSIP.664 trimer shown in Fig. 1d was selected as reference for comparison. The proportions of high-mannose and complex-type glycans at those glycosites highlighted in yellow were assigned based on the proportion of spectra hits since peak area did not reach the threshold of 5E8 (Supplementary Tables 24–33).

workflows typically provide little information about site occupancy[24,79], unless a separate analysis is conducted using PNGase F-treated Env peptides and screening is carried out for peptides with Asn versus Asp, as we have done here[28,31]. Indeed, analysis of site occupancy is a routine outcome of our method, and we have found that there is >90% site occupancy for all but 2–5 of 25–29 sites in each of the soluble Env trimers of 6 HIV/SIV strains examined. The one or more of the glycosites in the gp41 subunits of all 6 Env trimers except SIVcpzMT145 SOSIP.664 were not fully glycosylated, perhaps as a consequence of being released from the signal anchor before the oligosaccharyltransferase had a chance to act[80,81]. A number of glycosites on the BG505 gp120 monomer were also only partially glycosylated, presumably due to the different kinetics of synthesis and folding compared with the SOSIP trimer[81,82]. Beyond site occupancy, quantification of the glycoforms at each site by glycoproteomics is also a significant challenge due to the multiple glycoforms at each site, and the unknown efficiency for detection of peptides with each glycoform due to the impact of glycans with different number of sialic acid moieties on ionization efficiency during MS analysis[63,83]. Thus, detection of a glycoform at a given site provides evidence for the types of glycans found at that site, but without reliable means to routinely assess the relative abundance[27–31,79].

In a thorough glycoproteomics analysis of glycopeptide enriched fractions from a BG505 SOSIP.664 trimer, Berhens *et al.*[24,34] detected 20–26 of the 28 glycosites and summed peak areas for each high-mannose and complex-type glycoform detected as a measure of their abundance at each site. The high-mannose and complex-type glycoforms detected at each site were consistent with the ordered processing pathway that first trims the high-mannose structures from $Man_9$ to $Man_5$ before adding terminal sequences that define complex glycans. Thus, for sites with 100% high-mannose, there was mainly $Man_9$. For sites with a mixture of high-mannose and complex-type, there were mainly mixtures of processed high-mannose structures ($Man_8$ to $Man_5$) and simple complex-type structures[24]. Thus, as would be expected, defining the ratio of high-mannose to complex-type glycans, as we have done here, provides insight into the degree of processing that has occurred at each site. In this regard, it is notable that there is quite good concordance between the ratio of high-mannose versus complex-type glycans observed for all sites in the BG505 SOSIP trimers analysed here and by Berhrens *et al.*[24]. Thus, we suggest that the global analysis of site-specific glycosylation described here provides a great deal of information about access of each site to the processing machinery.

Given the importance of glycosylation on the immunogenicity and biological activity of HIV Env[20,21,84,85], we believe that this

semi-quantitative proteomics-based method fills a need to routinely get detailed information about site-specific glycosylation of Env. Indeed, the glycosylation of HIV-1 Env is potentially impacted by various factors, such as purification methods[23], batch-to-batch variation, cell type used for expression[24,25,31], expression as soluble or membrane form[25,28], structural constraints[24] and strain/genotype[26,27,30]. Using the native-like Env soluble trimers as a model, the method revealed little difference in site-specific glycosylation in different biological batches, or in the method of purification (Fig. 4), but significant differences in site-specific glycan processing across various strains of HIV (Fig. 5). Such information will also inform high resolution X-ray and cryo-EM structure analysis of Env where electron density for glycans can be visualized. In one recent report on the crystal structure of BG505 SOSIP.664 trimer, we note that glycans at sites N301 and N276 were assigned complex-type, while predominately high mannose type was seen here[86]. So clearly there will be opportunity for these methods to provide complementary information for refining Env structure analysis relevant for vaccine design. In this regard, it will be of immediate interest to further establish the degree to which glycan processing differs between soluble, membrane bound and viral forms of Env, and the degree to which glycosylation of viral Env matches that of recombinant Env trimers or Env fragments being considered as vaccine candidates.

Finally, although we have used HIV Env as a model, the method is suitable for analysis of site-specific glycosylation of any glycoprotein. It should be particularly useful for analysis of the glycosylation of cell surface glycoproteins, viral glycoproteins and soluble recombinant glycoproteins and could be an additional tool for analysis of biotherapeutics to assess in one-step site-specific occupancy and degree of glycan processing.

## Methods

**Expression and purification of Env trimers and gp120.** SOSIP.664 trimer[52] and gp120 constructs were transiently transfected using 293 Fectin (Invitrogen) into 293F cells, cultured in 293 Freestyle media (Life Technologies) and grown for 96–144 h. SOSIP.664 timers were co-transfected with furin plasmids. For the BG505 SOSIP.664 trimer produced with kifunensine (BG505_Kif), 15 uM of kifunensine (Cayman Chemical) was added to 293 Freestyle media. After harvest and sterile filtration on a 0.22 μM filter (Nalgene), the supernatants were purified using the following protocols:

*His-tag purification.* Unless indicated otherwise, all N-terminally His-tagged BG505 SOSIP.664 and BG505_Kif constructs were purified in two steps using a HIS-TRAP affinity column (GE Healthcare) with a linear elution gradient from 20 mM Imidazole to 500 mM Imidazole, followed by a Superdex 200 Increase SEC column (GE Healthcare) in Hank's Buffered salt solution (HBE) (10 mM HEPES 150 mM NaCl)[71]. The oligomeric state and purity were assessed by size exclusion chromatography-mutli-angle light scattering (Wyatt Technology DAWN HELEOS II, Wyatt Technology Optilab T-rEX).

*Antibody affinity—positive selection.* In this approach, N-terminally Avi-tagged B41 SOSIP.664 and 327c SOSIP.664, as well as untagged CRF02_AG_250 SOSIP.664 and SIV$_{cpz}$MT145 SOSIP.664 constructs were transfected as described above. Supernatants of B41 SOSIP.664 and 327c SOSIP.664 were subsequently purified over 2G12 antibody affinity column, while CRF02_AG_250 SOSIP.664 and SIV$_{cpz}$MT145 SOSIP.664 SOSIP.664 were purified by using PGT145 antibody affinity column. Supernatants were slowly run over a CNBr-activated Sepharose 4 Fast Flow (GE) conjugated antibody affinity column at 4 °C overnight, eluted in 3 M Magnesium chloride and buffer exchanged into TBS. The eluate was then run over a Superdex 200 Increase SEC column (GE Healthcare) in TBS and trimer-size fractions were collected. The trimeric state of all proteins was confirmed by blue-native PAGE.

*Antibody affinity - negative selection.* In this approach, untagged JR-FL SOSIP.664 trimers were purified by negative selection. The filtered culture supernatants were run over a Galanthus nivalis lectin (GNL) affinity chromatography column overnight at 4 °C to isolate the heavily glycosylated envelope glycoproteins. The eluate containing the JR-FL SOSIP protein was run over a Superdex 200 10/300 GL SEC column to separate trimer-size oligomers from aggregates and gp140 monomers. Trimer-size JR-FL SOSIP containing fractions were loaded onto a protein A agarose column previously loaded with two-fold weight excess of non-neutralizing mAb F105 over the amount of trimeric SOSIP loaded. The column was gently rocked at 4 °C for 45 min, the solid phase was allowed to settle for 5 min and the well-ordered trimeric JR-FL SOSIP protein was recovered from the flow-throw in a negative selection manner.

**Combination proteolytic digestion of Envs.** The proteins (∼50 μg) were denatured with 8 M urea in 100 μl of 100 mM ammonium acetate (pH 6) to minimize nonenzymatic deamidation[87]. The denatured proteins were reduced with 10 mM of dithiothreitol (DTT) at 56 °C for 1 h, followed by iodoacetamide treatment (50 mM, 45 min, in the dark) to alkylate cysteine residues. The reduced and alkylated samples were buffer exchanged using 10 kDa centrifugal filters (Millipore). The resulting proteins dissolved in 100 mM ammonium bicarbonate (pH 8) were divided into five equal aliquots for the following proteolytic digestion.

*Triple digestion.* Three out of five aliquots were digested with different proteases by using a modified triple digestion protocol, described previously[42]. Arg-C followed by trypsin. One fraction was first digested with Arg-C (Promega) at an enzyme/substrate ratio of 1:20 (w/w) in 100 μl of 100 mM ammonium bicarbonate (pH 8) containing 5 mM DTT and 0.2 mM EDTA. The reaction solution was placed at 37 °C for 4 h. The resulting sample was lyophilized to remove volatile salt ammonium bicarbonate, and was re-dissolved in 500 μl of 100 mM ammonium acetate (pH 6) subsequently. Sequencing grade modified trypsin (Promega) was added to the solution at an enzyme/substrate ratio of 1:10 (w/w). The protein was digested at 37 °C for 16 h. After incubation, the sample was lyophilized. 'Elastase'. A second fraction was digested with elastase (Promega) at an enzyme/substrate ratio of 1:20 (w/w) in 500 μl of 100 mM ammonium bicarbonate (pH 8). The reaction was incubated at 37 °C for 16 h. After incubation, the sample was lyophilized. 'Subtilisin'. A third fraction was digested with subtilisin (Sigma) at an enzyme/substrate ratio of 1:20 (w/w) in 500 μl of 100 mM ammonium bicarbonate (pH 8). The digestion was conducted at 37 °C for 4 h. After incubation, the sample was lyophilized. Three fractions were then re-dissolved in water and combined into one microcentrifuge tube, and the protease enzymes were denatured at 100 °C for 5 min.

*Chymotrypsin.* A fourth fraction was digested with chymotrypsin (Promega) at an enzyme/substrate ratio of 1:13 (w/w) in 500 μl of 100 mM ammonium bicarbonate (pH 8). The reaction was conducted in a 30 °C water bath for 10 h. The reaction buffer was lyophilized, and the peptides were re-dissolved in water. The protease enzyme was denatured at 100 °C for 5 min.

*Trypsin and chymotrypsin.* The fifth was digested with the combination of trypsin and chymotrypsin at an enzyme/substrate ratios of 1:20 (w/w) and 1:13 (w/w) respectively in 500 μl of 100 mM ammonium bicarbonate (pH 8). The reaction was conducted at 37 °C for 16 h. The reaction buffer was lyophilized, and the peptides were re-dissolved in water. The protease enzymes were denatured at 100 °C for 5 min.

**Deglycosylation.** The samples generated from different proteolytic digestions were separately de-glycosylated with Endo H, followed by PNGase F treatment.

*Endo H.* Endo H (New England Biolabs) was added to the (glyco)peptides (250 units per 10 μg) re-dissolved in 20 μl of 100 mM ammonium acetate (pH 5.5). The reaction was incubated at 37 °C for 1 h.

*PNGase F.* The Endo H-treated (glyco)peptides were lyophilized and then re-dissolved in 100 mM ammonium bicarbonate (pH 8) prepared with $O^{18}$-$H_2O$ (97%, Sigma). PNGase F (lyophilized, New England Biolabs) re-dissolved in $O^{18}$-$H_2O$ (97%, Sigma) was added to the (glycol)peptide solution (500 units/10 μg). The reaction was allowed to proceed at 37 °C for 1 h in a sealed microcentrifuge tube. The endoglycosidase enzymes were denatured at 100 °C for 5 min.

**Endoglycosidase validation experiment using BG505_Kif.** BG505_Kif was denatured, reduced, and alkylated as described above. The protein solution was then divided into three aliquots, which were digested with triple digestion, chymotrypsin, and the combination of trypsin and chymotrypsin, respectively. The reaction buffer was lyophilized, and the peptides were re-dissolved in water. The protease enzymes were denatured at 100 °C for 5 min and lyophilized. The resulting peptides generated from each proteolytic digestion were re-dissolved in 40 μl of 100 mM ammonium acetate (pH 5.5), and then divided into two equal aliquots with equal volume. One aliquot was 'mock digested' with 100 mM ammonium acetate (pH 5.5) buffer only (no Endo H), followed by PNGase F treatment. Another one was de-glycosylated with sequential Endo H and PNGase F treatments. The resulting samples were either directly analysed to validate endoglycosidase treatments or equal volumes were mixed to generate a sample with N + 3 and N + 203 modified peptides at a molar ratio of 1:1 for characterizing MS detection of glycotypes.

**Mass spectrometric analysis.** The samples from HIV-1/SIV Envs and the recombinant influenza virus haemagglutinin were analysed on a Fusion Orbitrap tribrid mass spectrometer (Thermo Fisher Scientific). Approximately 1 ug of de-glycosylated peptides were injected directly onto a 30 cm, 75 um ID column packed with BEH 1.7 μm C18 resin (Waters). Samples were separated at a flow rate of 200 nl min$^{-1}$ on a nLC 1000 (Thermo Fisher Scientific). Buffer A and B were 0.1% formic acid in water and acetonitrile, respectively. A 240-minute gradient was run, consisting of the following steps: 0–150 min, 5–22% B; 150–200 min, 22–32% B; 200–210 min, 32–90% B and hold at 90% B for a final 30 min of run time.

Column was re-equilibrated with 20 μl of buffer A prior to the injection of sample. Peptides were eluted directly from the tip of the column and nanosprayed directly into the mass spectrometer by application of 2.5 kV voltage at the back of the column. The Orbitrap Fusion was operated in a data-dependent mode. Full MS1 scans were collected in the Orbitrap at 120 K resolution with a mass range of 350–1,500 $m/z$ and an AGC target of 4e (ref. 5). The cycle time was set to 3 s, and within this 3 s the most abundant ions per scan were selected for CID MS/MS in the orbitrap with an AGC target of 5e (ref. 4) and minimum intensity of 5,000. Maximum fill times were set to 50 and 100 ms for MS and MS/MS scans, respectively. Quadrupole isolation at 1.6 $m/z$ was used, monoisotopic precursor selection was enabled and dynamic exclusion was used with exclusion duration of 5 s.

Bovine fetuin was separately digested with triple digestion, chymotrypsin, and the mixture of trypsin and chymotrypsin. The peptides that were generated from three digestions were combined at a molar ratio of 1:1:1 after deglycosylation. Approximately 2 ug of the peptides were pressure-loaded onto a 100 μm i.d fused silica capillary MudPIT column containing 1.5 cm of 10 μm Jupiter C18 (Phenomenex) followed by 1.5 cm of 5 μm Partisphere strong cation exchange (SCX, Whatman). The MudPIT column was connected to a 75 μm capillary separation column packed with 15 cm of 4 μm Jupiter C18 (Phenomenex) and was attached to an Agilent 1200 quaternary HPLC placed in line with the heated capillary of an LTQ Orbitrap XL Mass Spectrometer (Thermo Fisher Scientific). The buffers used were A: 5% acetonitrile/0.1% formic acid, B: 80% acetonitrile/0.1% formic acid, and C: 500 mM ammonium acetate in 5% acetonitrile/0.1% formic acid. Peptides were separated using a nine-step separation. The first step was 59 min with the following gradient: 0–5 min, 0–15% B; 5–35 min, 15–70% B; 35–40 min, 70–100% B; 100% B, 40–45 min; 100–0% B, 45–46 min; 0% B, 46–59 min. The next eight steps were 120 min each with the following gradient: 100% A, 0–4.9 min; 5–8 min, x% C; 100% A, 8.1–10 min; 100–85% A, 10–12 min; 85–50% A, 12–100 min; 50–0% A, 100–105 min; 0% A, 105–110 min; 100% A, 110.1–120 min. The 3 min buffer C percentage (x) in steps 2–9 were as follows: 20, 30, 40, 50, 60, 70, 90 and 100. As peptides were eluted from the microcapillary column, they were ionized by the application of a distal 2.5 kV spray voltage. Ionized peptides were detected with one survey scan (300–2,000 $m/z$) in the Orbitrap with nominal resolution of 60,000 followed by 10 data-dependent MS/MS scans at 35% normalized collision energy. Charge state rejection was set to omit singly charged ion species for MS/MS but to include ions for which a charge state could not be determined. Dynamic exclusion was enabled with a repeat count: 1, duration: 30.0, list size: 500, exclusion duration 120.0, exclusion mass with high/low: 1.51/0.51 $m/z$, early expiration enabled.

**Data processing.** MS and MS/MS data were extracted from RAW files by using RawConverter[46], enabling monoisotopic correction. The MS/MS spectra were searched using the ProLuCID algorithm from software package Integrated Proteomics Pipeline-IP2 (version 4)[43–45,47] against the European Bioinformatic Institute (IPI) *Bos taurus* protein database including the sequences of HIV-1/SIV Envs analysed in this study. A target/decoy database containing the reversed sequences of all the proteins appended to the target database was used so that we could estimate peptide probabilities and FDRs accurately. MS tolerances were set at 50 p.p.m. for precursor ions and 20 p.p.m. (was 600 p.p.m. for the bovine fetuin data) for fragment ions. No enzyme specificity was considered for the searching. Carboxyamidomethylation ($+57.02146$ C) was considered as a fixed modification. In addition, oxidation ($+15.9994$ M), deamidation ($+2.988261$ N), GlcNAc ($+203.079373$ N), and pyroglutamate formation from N-terminal glutamine residue ($-17.026549$ Q) were searched as variable modifications. The results were filtered by DTASelect (version 2.0). The parameters were set as: minimum number of peptide per protein $\geq 2$, spectrum false positive rate $\leq 0.05$, and precursor delta mass cutoff $\leq 10$ p.p.m. (was 20 p.p.m. for the bovine fetuin data). The remaining peptides were further filtered to remove those peptides with N + 3 and N + 203 modifications that were not located at the consensus motif (N-X-S/T, X can be any amino acid residue except proline). We used Census[47] to reconstruct chromatograms for identified peptides. Each peak was smoothed and fit to Gaussian distribution to calculate the abundance of peptide using peak area. Peak area was further normalized by using ion injection time. We aligned spectra from different samples by using precursor mass tolerance and retention time to calculate relative ratios.

**Statistical analysis.** The following statistical analyses were performed with GraphPad Prism 5. The ion intensity peak area of the peptides from each raw file was generated by summing the peak area of this peptide over all identified charge states[24], and that was used for statistical analysis. A set of peptides with N + 0, N + 3 and N + 203 modifications was chosen only when at least one out of the three was identified with peak area more than 5E8. If no peptide for a given glycosite met the threshold of 5E8, proportions of the three glycosylation states at this site were generated based on spectra hits[28]. BG505 SOSIP.664 trimer purified with $Ni^{2+}$/SEC that was shown in Fig. 1d was selected as reference, to which proportions of the three glycosylation states at each glycosite on other Envs analysed in this study were compared by using a Mann–Whitney test. Significant differences were concluded only when the $P$ value was $<0.05$ and the difference on the proportion of no glycan, high-mannose, or complex-type glycans was $>10\%$.

**Data availability.** The data that support the finding of this study can be downloaded from the MassIVE site (http://massive.ucsd.edu/ProteoSAFe/data-set.jsp?task=05eced07b0f9486fb04509a4aec1197a) or available from the corresponding author upon reasonable request.

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

## Acknowledgements

We thank Dr Ian A. Wilson and Dr Peter S. Lee for providing recombinant influenza virus haemagglutinin. This work was supported by NIH R01AI113867 (J.C.P., J.Y., W.R.S.); NIH UM1 AI100663 (D.R.B.); the International AIDS Vaccine Initiative (W.R.S., D.R.B.); and NIH P41 GM103533 (J.Y.).

## Author contributions

L.C., J.R.Y. and J.C.P. designed the research. L.C. prepared samples for MS analysis. J.K.D., L.C. and C.M.D. performed the MS analysis. L.C., L.H. and S.R.P. analysed the data. D.W.K., M.M., D.S., C.Y.S., S.M., R.A., J.G., E.G., M.K. and Y.A. expressed and purified Env proteins. D.R.B., W.R.S., J.R.Y. and J.C.P. supervised the project. L.C. and J.C.P. wrote the manuscript.

## Additional information

**Competing interests:** The authors declare no competing financial interests.

