## [Peer Review File · Nature Communications]

Reviewers' Comments:

Reviewer #1 (Remarks to the Author)

Paulson and co-workers report a site-specific analysis of HIV glycosylation for N-glycosites. HIV Env trimer is a large, highly glycosylated protein that has been previously studied but without complete sequence coverage. The authors rely on the differential recognition of PNGase F and EndoH for cleaving different types of glycans, and with the assistance of multiple proteases, are able to sequence >95% of Env including all glycosites. This is a creative application of these enzymes to gain additional distinction between high mannose and complex N-glycans from a sample, and would be of interest to the community seeking to rapidly identify the degree of complexity of N-glycosylation at a particular site. Drawbacks in the methods include the reliance on PNGase F and Endo H enzymes and the remaining inability to distinguish hybrid versus complex (or degree of complexity) glycosylation.

- The title is quite general and should be made more specific for the glycoprotein and the type of glycosylation (N-glycans).
- The authors comment that quantification only occurs at 5E8. What about any peptides that are present, but below 5E8? Is there differentiation between present, but not quantified?
- There is the analytical challenge of quantifying the same glycosite from different peptides across treatment types. Particularly when comparing glycosylated versus nonglycosylated peptide ionization efficiency. How do the authors actually get around this challenge?
- An additional comparison of PNGase F and EndoH with high mannose protein would be a useful contrast to the data from fetuin.
- Line 374-378: The authors note that converting the glycomics to proteomics problem increases their identification rate and time. This is true, but this is also essentially what all other labs have done to address the glycoproteomics problem with N-glycans. Their addition here is to get one more facet of distinction between complex and high mannose glycans, as compared to glycosylated versus not glycosylated. True glycoproteomics efforts, which have both the glycomic and proteomic problem, are indeed analytically challenging, but do gain additional information about specific glycan structures and their peptide substrates. The language therefore should be placed within this perspective that the vast majority of glycoproteomics efforts in the literature employ PNGase F and would be able to be performed in the same time frame as the study herein.

Reviewer #2 (Remarks to the Author)

The paper by Cao et al., “Global site-specific analysis of HIV glycosylation” describes an analytical approach for glycomics which is presented to be “a general mass spectrometry-based proteomics strategy that uses specific endoglycosidases to introduce mass signatures that distinguish peptide glycosites that are unoccupied or occupied by high-mannose or complex-type glycans”. The authors conclude, “[the method] should be particularly useful for analysis of the glycosylation of cell surface glycoproteins, viral glycoproteins and soluble recombinant glycoproteins and could potentially provide the basis for routine monitoring of the glycosylation of biotherapeutics.”

The paper will be of interest to scientist interested in glycosylation and particularly to investigators within the HIV glycosylation field. The potential to apply this method to biotherapeutics is interesting, although the approach deliberately throws out details of the glycosylation structures. It is necessary to define these structures for regulatory approval so this method would not generate sufficient information for such applications. In contrast, this method may find application in the monitoring of glycosylation for batch-to-batch variation or, for example, during continuous feed bioreactors.

The main limitation of the method, as far as publication in a top-tier journal is concerned, is lack of novelty. The method is highly redundant with many published methods in specialised journals and the current manuscript really only is novel insofar as the method has been executed on an interesting body of samples.

Example papers disclosing the use of endoglycosidases in proteomics:

In the HIV glycosylation papers: “Methods development for analysis of partially deglycosylated proteins and application to an HIV envelope protein vaccine candidate” by Go and “Characterization of glycosylation profiles of HIV-1 transmitted/founder envelopes by mass spectrometry” by Go, the earlier paper by Hagglund is cited describing the type of approach using endoglycosidase and endopeptidases in detail:

An enzymatic deglycosylation scheme enabling identification of core fucosylated N-glycans and O-glycosylation site mapping of human plasma proteins Hagglund P., Matthiesen R., Elortza F., Hojrup P., Roepstorff P., Jensen O.N., Bunkenborg J. (2007) *Journal of Proteome Research*, 6 (8) , pp. 3021-3031.

The O18 method has been used extensively e.g.

“N-Glycosidase treatment with 18O labeling and de novo sequencing argues for flagellin FliC glycopolymerism in *Pseudomonas aeruginosa*”

“Determination of site-specific glycan heterogeneity on glycoproteins” by Kolarich

“Development of a combined chemical and enzymatic approach for the mass spectrometric

identification and quantification of aberrant N-glycosylation”

Angel et al., 2006 Rapid Commun in Mass Spec; Gonzales et al., 1992 Analytical Biochemistry; Haeggung et al., 2007 Journal of Proteome Research; Kaji et al., 2003 Nature Biotechnology
Angel et al., above highlights important problems regarding this workflow based on residual trypsin activity, which are also not discussed or addressed in the present manuscript. This establishes an unknown limitation on the described data.

Specific comments:

Introduction

- Line 62ff – references 23 and 24 may be mixed here. Given the ref title, I assume Pritchard et al, should be the reference for the structural constraints.
- Line 73: not all complex glycans are terminated with sialic acids
- Line 94: while the previous paragraph described the workflow for the Behrens et al., paper – the Go et al., references here do not really fit as their workflow is not restricted by the creation of glycan libraries or glycan enrichment. Also the argument about occupancy doesn't fit for the Go papers, as in Go papers they do not enrich for glycopeptides and provide occupancy information in some of their papers.

Results

- Ll 114 to 126: refs are missing. This section sounds as if the authors invented the here described workflow (see above).
- Line 181: “previous analysis revealed that recombinant BG505 SOSIP.664 is very low in hybrid glycan content”. This is not entirely true and also not consistent with the cited references. First of all, Panico et al., did not analyse BG505 SOSIP.664. They also indeed identified hybrid type glycans on two N-glycan sites. Behrens et al., paper identified hybrids on some N-sites. Hence, the argument that hybrids can be ignored in this workflow is not suitable and marks a downside of the workflow, especially if it should be transferred to further targets.
- L. 260 – How was the molar ratio of 1:1 verified as this is very difficult to do/control using unlabelled peptides. No details in the method section.
- L.310 ff: As this workflow would only detect changes from oligomannose to complex or the other way around, but not changes to e.g. smaller oligomannose type structures or smaller complex structures – it can't really be concluded that there is no significant effect on processing neighbouring glycans
- Ll. 332 ff Are all of these trimers of the SOSIP design? Why are not all of them His tagged for consistency (B41 is not)? It should be commented on how they were produced and purified. Judging from the Method section, all of them were purified differently indicating that the experimental design has not controlled all variables.

Discussion

- Ll. 392: the here discussed negative aspects of glycopeptide quantitation is not balanced as

there are also a number of papers arguing that the actually glycan present on a glycopeptide does not impact its ionization efficiency e.g. Wada 2007 Glycobiology

- LL 399 Panico et al., did not construct a glycan library per se. They just also looked at the glycomics of the sample without per se using this data as a library. Hence “several workflows” is a bit exaggerated.

Figures/Tables:

Figure 1C:

It is not entirely clear, what exactly each dot represents – replicates/same site on different peptides – both of it?

SI Figure 1: Why are sites 99 and 156 represented by 7 dots (replicates?) vs just one dot for the third site (176)

SI Figure 5: Construct names are misleading, if all of them are of the SOSIP design, add SOSIP to all names. E.g. B41 SOSIP.664

Conclusion

The manuscript presents interesting and useful information about Env glycosylation, particularly the occupancy which has only been investigated by Go et al on other Env variants. The workflow uses established techniques and the paper should probably be judged on the biological insights presented. These include occupancy measurements of the glycan sites and comparisons with strains that go well beyond Behrens. The paper will therefore be of considerable interest to those working in that area.

Reviewer #3 (Remarks to the Author)

Cao et al. described a novel method for characterization of protein glycosylation with precise and quantitative mapping of site occupancy, and demonstrated its utility on HIV-1 envelope glycoprotein, the key protein contributing to virulence of HIV and the main target of neutralization antibodies for HIV management. The method cleverly combines a series of existing methods, including O18 labeling, endoglycosidases of endo H and PNGase F, and different proteases, in defined order to quantitatively characterize the site-specific occupancy for three forms of N-linked glycosylation: complex, high mannose (hybrid), and no glycans. The information is critical for functional studies of these glycans and their carrier proteins, which are not limited to HIV envelope glycoproteins, but to other glycoproteins encoded by a large portion of the genome regardless of the species.

The method is described in detail and thoroughly characterized particularly with the efficiency of endoglycosidases. The references are sufficient, and the writing can be improved in a few places listed below.

1) The study used three different proteolysis methods for HIV-1 Env protein and each separately processed by endo H and PNGase F and finally analyzed by LC-MS in triplicates. So 9 samples are characterized by LC-MS from each HIV-1 Env protein and with 3 technical replicates, which is a careful analysis design. The biological replicates were achieved by comparing the result of regular Env trimer with those experienced different enrichment approaches and those expressed with inhibitor, which is an efficient design to validate robustness of the method and to maximize the obtained information at the same time.

There is no comment on the reason of choosing the specific combination of three different proteolysis methods. It would be helpful for future users to know whether the use of chymotrypsin, trypsin and chymotrypsin, and ‘triple digestion’ is a must or the parallel proteolysis can be simplified to two instead of three routes.

2) The study employed different combinations of proteases for mapping the glycosylation states. On page 7, “peptide sets” is used to describe the results of three types of possible glycosylation, i.e. high mannose, complex, and no-glycans. But it is unclear to me what protease results they represent. Is it possible to have multiple peptide sets at one particular glycosylation site in a single proteolysis scheme? If so, is this due to miss cleavage?

3) The use of different protease in combination is one novelty for mapping the glycosylation occupancy. The observed biases in some “peptide sets” mentioned on page 7, lines 187 and 188 can be included in supplementary with a breakdown on the used proteases. A further explanation of the apparent biases would be helpful. Will it be possible that certain proteases have preferences to cleave peptides with certain glycan structure, or their presence is stochastic?

4) The method used the selectivity of endo H and PNGase F to distinguish the types of N-linked glycans, in which the high mannose structure recognized by endo H does not exclude the hybrid glycans that include both high mannose and complex glycans. This concern was discussed on page 7, line 178-181 for HIV-1 Env. It is also helpful to mention this confounding effect in the introduction and in the discussion when the method was recommended to the characterization of general glycoproteins.

In summary, it is a solid study and addresses an important area that concerned by a number of fields including but not limited to virology, immunotherapy, glycobiology, and signaling. The study is suitable to publish in Nature Communication with additional edition.

Responses to Reviewer comments:

Reviewer #1:

Paulson and co-workers report a site-specific analysis of HIV glycosylation for N-glycosites. HIV Env trimer is a large, highly glycosylated protein that has been previously studied but without complete sequence coverage. The authors rely on the differential recognition of PNGase F and EndoH for cleaving different types of glycans, and with the assistance of multiple proteases, are able to sequence >95% of Env including all glycosites. This is a creative application of these enzymes to gain additional distinction between high mannose and complex N-glycans from a sample, and would be of interest to the community seeking to rapidly identify the degree of complexity of N-glycosylation at a particular site.

We thank the reviewer for these positive comments about the method.

Drawbacks in the methods include the reliance on PNGase F and Endo H enzymes and the remaining inability to distinguish hybrid versus complex (or degree of complexity) glycosylation.

We assume that the reliance on PNGase F and Endo H is considered a drawback because information gained about nature of the glycans at each site relies on the known specificity of these enzymes, and belies the true complexity of the glycoforms that are there. We agree, but have embraced this limitation as a tradeoff for enabling the analysis of the glycosylation status of every glycosite, and reducing the time required for analysis from a few months to a few days. In effect, by relying on the specificity of the enzymes for probing the glycosylation status, the analytical challenge shifts from a glycomics problem to a very manageable proteomics problem. This approach has allowed us to apply standard database searching and label free analysis to conduct high-throughput identification and quantification of peptides representing the three-glycosylation states of each glycosite.

We agree that the current method is not able to distinguish hybrid from oligomannose glycosylation since Endo H release both high mannose and hybrid glycans, and these structure types are viewed as one class. We have further emphasized this fact with statements in the introduction and in the discussion (Line 103-106, 126, and 395). While it would indeed be desirable to quantify hybrid type glycans as a separate class, the groups of Crispin and Dell have found that hybrid structures represent a small proportion of the total glycans (<5%).

- The title is quite general and should be made more specific for the glycoprotein and the type of glycosylation (N-glycans).

We agree that the manuscript is highly focused on HIV Env glycosylation, and have changed the title to a more specific one, namely “Global site-specific N-glycosylation analysis of HIV envelope glycoprotein”. That said, however, the method is general to any glycoprotein, and we believe that it will have broad utility outside the HIV field.

- The authors comment that quantification only occurs at 5E8. What about any peptides that are present, but below 5E8? Is there differentiation between present, but not quantified?

This is a great question. On the one hand it is an arbitrary threshold, but one that was carefully selected as optimal for including data that have utility for semi-quantitative analysis. We chose this threshold based on the control experiment by using BG505 SOSIP.664 trimer expressed with kifunensine (Kif_BG505). Thus Kif_BG505 should be occupied by only high mannose

glycans as kifunensine is an inhibitor of the mannosidase I enzyme. When Kif_BG505 was treated with only PNGase F, no Endo H treatment, we should get homogeneous (N+3) signature for all occupied glycosites on Kif_BG505 (Fig 3c). Indeed the proportion of (N+3) peaks (purple bars) is 98% in total of N+3 and N+203 peaks when the threshold is 5E8. Low percentages of N+203 appearing at several glycosites (2% in total) are likely false positives generated from data processing. But the proportion of false positives (green bars) increased to 4% when the threshold of peak area was reduced from 5E8 to 5E7 (Supplementary Figure 5). We lost quantitative data at some glycosites if the threshold was further increased to 5E9 (data not shown). Therefore, we set the threshold of peak area at 5E8 in this paper. We suggest people who want to use this method should choose this threshold by themselves based on control experiments as the threshold may vary with the type of mass spectrometer used.

For most glycosites many distinct peptides are detected (up to 43) due to the nature of the multiple protease treatments. We include the peptide in the final results if one of the three forms in the set (+0, +203, +3) meets the threshold of 5E8. This provides us with a more reliable analysis of the proportion of the three forms/glycosylation states at a specific glycosite. The threshold of 5E8 also allowed us to see all glycosites of Env across different HIV-1/SIV strains. Peptides that are present, but below 5E8 have low spectrum and peptide FDR (less than 1%), and are just omitted from the analysis to avoid errors due to undersampling.

The impact of choosing 5E8 vs 5E7 is seen by comparing Fig 3c and supplementary Fig 5, which were generated from the same data, but using the different peak thresholds (5E8 and 5E7, respectively). The results are basically same, but as described above, the proportion of false positives will increase slightly when you reduce the peak area threshold.

- There is the analytical challenge of quantifying the same glycosite from different peptides across treatment types. Particularly when comparing glycosylated versus nonglycosylated peptide ionization efficiency. How do the authors actually get around this challenge?

This is a key point. We believe that data obtained by us and others suggest that it will not a major problem for the vast majority of peptides. We started with the assumption that there would not be major differences in ionization efficiencies based on the careful work of Kolarich et al. (Daniel Kolarich et al., J. Mass Spectrom. 2013, 48, 627–639), who analyzed matched sets of synthetic peptides containing N-glycan sequons with Asn (+0), Asp (+3), or GlcNAc-Asn (+203) and measured their detection efficiency in MS/MS experiments. They found only modest differences ($\pm 20\%$) in detection efficiency. We then conducted our own experiment, which is highlighted in Figure 3. We produced a single batch of BG505 Env in the presence of kifunensine, which produces proteins with only high mannose glycans. When treated with Endo H followed by PNGase F, all glycosites are converted to GlcNAc-Asn (+203), regardless of what they are naturally. When treated instead with PNGase only, all glycosites are converted to Asp (+3). Then we mixed the digestions 1:1, where effectively the +203 and +3 versions of every glycosite would be present in a 1:1 ratio. The actual ratios detected at each site varied somewhat from site to site (See Fig. 3d), but the average was only slightly in favor of the Asp (+3) with a ratio of 1:1.2. More importantly, we asked if the results for glycosylation of BG505 (Figure 1) would change if we used the ratio calculated for each individual glycosite as the most extreme way of correcting the data for ionization efficiencies. The result was that there was only one site (at 637) that differed significantly with or without the correction (See Supplementary Figure 6). From this point on we decided not to apply a correction. While we acknowledge that there might be a rare case that an aberrant result could be obtained for a given peptide as a result of ionization efficiency differences, we are confident it is not a major problem.

- An additional comparison of PNGase F and Endo H with high mannose protein would be a useful contrast to the data from fetuin.

We thank the reviewer for the suggestion. Effectively the BG505 protein produced in Kifunesine is also a high mannose protein that tests the system (Fig. 3b). However, in response to the suggestion to show that a natural protein containing high mannose complements the fetuin control, we added a figure to supplementary information generated from invertase a glycoprotein with high mannose glycans produced by the yeast *S. cerevisiae* (Supplementary Fig. 2).

- Line 374-378: The authors note that converting the glycomics to proteomics problem increases their identification rate and time. This is true, but this is also essentially what all other labs have done to address the glycoproteomics problem with N-glycans. Their addition here is to get one more facet of distinction between complex and high mannose glycans, as compared to glycosylated versus not glycosylated. True glycoproteomics efforts, which have both the glycomic and proteomic problem, are indeed analytically challenging, but do gain additional information about specific glycan structures and their peptide substrates. The language therefore should be placed within this perspective that the vast majority of glycoproteomics efforts in the literature employ PNGase F and would be able to be performed in the same time frame as the study herein.

We agree that PNGase is routinely used in the proteomics field to identify glycosites, and Endo H is also used effectively in some contexts, and that the time frame for such analysis is similar to our workflow. As pointed out by the referee, by using the PNGase and EndoH sequentially in our workflow, it has laid the foundation to semi-quantitative assessment of three different states of glycosylation. We had compared our workflow to those involving detailed analysis of glycans at each site, and the savings in time to get information at each glycosylation site. We have now taken care in the first paragraph of discussion to place our comments into more appropriate context with existing glyco-proteomics work (see also response to a similar comment from Reviewer 2 below).

Reviewer #2 (Remarks to the Author):

The paper by Cao et al., “Global site-specific analysis of HIV glycosylation” describes an analytical approach for glycomics which is presented to be “a general mass spectrometry-based proteomics strategy that uses specific endoglycosidases to introduce mass signatures that distinguish peptide glycosites that are unoccupied or occupied by high-mannose or complex-type glycans”. The authors conclude, “[the method] should be particularly useful for analysis of the glycosylation of cell surface glycoproteins, viral glycoproteins and soluble recombinant glycoproteins and could potentially provide the basis for routine monitoring of the glycosylation of biotherapeutics.”

The paper will be of interest to scientist interested in glycosylation and particularly to investigators within the HIV glycosylation field.

We thank the reviewer for these positive comments.

The potential to apply this method to biotherapeutics is interesting, although the approach deliberately throws out details of the glycosylation structures. It is necessary to define these structures for regulatory approval so this method would not generate sufficient information for such applications. In contrast, this method may find application in the monitoring of glycosylation for batch-to-batch variation or, for example, during continuous feed bioreactors.

This is a good point, glycomics analysis of biotherapeutic proteins is routine. Typically, it is analysis of the glycoforms on the entire protein, rather than site specific analysis. We have accordingly modified this sentence saying that this method could be an additional tool for biotherapeutics to assess in one-step site specific occupancy and degree of glycan processing.

This could be valuable for glycoproteins with multiple glycans that are not the same at each site (eg. Like the influenza virus hemagglutinin shown in Fig. S2).

The main limitation of the method, as far as publication in a top-tier journal is concerned, is lack of novelty. The method is highly redundant with many published methods in specialised journals and the current manuscript really only is novel insofar as the method has been executed on an interesting body of samples.

Example papers disclosing the use of endoglycosidases in proteomics:

In the HIV glycosylation papers: “Methods development for analysis of partially deglycosylated proteins and application to an HIV envelope protein vaccine candidate” by Go and “Characterization of glycosylation profiles of HIV-1 transmitted/founder envelopes by mass spectrometry” by Go, the earlier paper by Hagglund is cited describing the type of approach using endoglycosidase and endopeptidases in detail:

An enzymatic deglycosylation scheme enabling identification of core fucosylated N-glycans and O-glycosylation site mapping of human plasma proteins Hagglund P., Matthiesen R., Elortza F., Hojrup P., Roepstorff P., Jensen O.N., Bunkenborg J. (2007) Journal of Proteome Research, 6 (8) , pp. 3021-3031.

The O18 method has been used extensively e.g.

-“N-Glycosidase treatment with 18O labeling and de novo sequencing argues for flagellin FliC glycopolymerism in Pseudomonas aeruginosa”

-“Determination of site-specific glycan heterogeneity on glycoproteins” by Kolarich

-“Development of a combined chemical and enzymatic approach for the mass spectrometric identification and quantification of aberrant N-glycosylation”

-Angel et al., 2006 Rapid Commun in Mass Spec; Gonzales et al., 1992 Analytical Biochemistry; Haegglung et al., 2007 Journal of Proteome Research; Kaji et al., 2003 Nature Biotechnology

As nicely summarized by reviewer 2, and also commented on by reviewer 1, endoglycosidases are used by many laboratories in the proteomics field. This is particularly the case for PNGase F, which has been used to deduce utilization of Asn-X-Ser/Thr sequons, and using the specificity of Endo H/Endo D to establish high mannose and core fucosylated glycans at specific sites. It was not our intent to claim priority for the use of the endoglycosidases in glycoproteomics workflows.

To date no other report has used Endo H and PNGase sequentially, a simple permutation in the use of these enzymes that in our method provides the foundation for establishing mass signatures for semi-quantitative analysis of: 1) the degree of site occupancy and proportion of 2) oligomannose/hybrid and 3) complex type glycans in a single sample. The sequential use of the enzymes requires that the reactions each go to completion, which we have documented. Equally important, but less obvious, to the success of the method is the protease workflow, which increased our sequence coverage to 97%, with ion intensities of peptides at each glycosite sufficient to use peak area instead of spectral hits as a basis for quantitation. These features together resulted in a transformative semi-quantitative method for assessing the three states of glycosylation for each glycosite in a highly reproducible manner.

We have now revised the first paragraph of the discussion to better place the advantages of this method into context with the existing use of endoglycosidases in the proteomics/glycomics field.

Angel et al., above highlights important problems regarding this workflow based on residual trypsin activity, which are also not discussed or addressed in the present manuscript. This establishes an unknown limitation on the described data.

We agree this is an important point, and initially we were indeed concerned about incorporation of O¹⁸ into C-terminus of peptides during deglycosylation due to the presence of residual proteases. In the report by Angel et al., a high ratio of enzyme/substrate (1:1) was used increasing the potential for residual protease. We used a much lower enzyme/substrate ratio in this study, ranging from 1:10 to 1:20. More importantly, we heat denatured all protease enzymes before conducting deglycosylation steps (Line 548, 553, and 558). Finally, we checked for evidence of C-terminal O¹⁸, by searching for +2 Da to +4 Da were as differential/variable modifications of the C-terminal amino acid, and found that there could be no more than 5% of the peptides with O18 incorporation. We have now commented on this in the manuscript (Line 141-142).

Specific comments:

Introduction

• Line 62ff – references 23 and 24 may be mixed here. Given the ref title, I assume Pritchard et al, should be the reference for the structural constraints.

Indeed reference 23 (Pritchard et al.) is the reference for the structural constraints, but they also assessed the impact of different purification methods on glycosylation of HIV-1 Env in their study (Table 1 and Table S2), thus, we also cited it for the purification methods.

• Line 73: not all complex glycans are terminated with sialic acids

Thank you, we agree, and this sentence is revised to better describe optional glycoforms (Line 74 to 75).

• Line 94: while the previous paragraph described the workflow for the Behrens et al., paper – the Go et al., references here do not really fit as their workflow is not restricted by the creation of glycan libraries or glycan enrichment. Also the argument about occupancy doesn't fit for the Go papers, as in Go papers they do not enrich for glycopeptides and provide occupancy information in some of their papers.

Thank you, yes that paragraph was specific for the Behrens et al paper, and we have deleted the Go papers, which are also appropriately cited above.

Results

• LI 114 to 126: refs are missing. This section sounds as if the authors invented the here described workflow (see above).

We have modified these introductory sentences to better emphasize the sequential treatment of PNGase F and Endo H, that are unique to our workflow, while placing it into context with previous work. (Line 118 to 125).

• Line 181: “previous analysis revealed that recombinant BG505 SOSIP.664 is very low in hybrid glycan content”. This is not entirely true and also not consistent with the cited references. First of all, Panico et al., did not analyse BG505 SOSIP.664. They also indeed identified hybrid type glycans on two N-glycan sites. Behrens et al., paper identified hybrids on some N-sites.

Hence, the argument that hybrids can be ignored in this workflow is not suitable and marks a downside of the workflow, especially if it should be transferred to further targets.

This is helpful. We agree that the Behrens paper is the most relevant here since that is what we are analyzing, and have removed the Panico reference. We have reframed the sentence to simply point out that hybrids will be included in the high mannose category. Actually Behrens's paper is very clear that hybrids can barely be detected in the pooled glycans, but reach 10-20% for a few of the 20 glycosites analyzed (197, 355, 637). We do not want to minimize the importance of hybrid or any other glycoforms, our main goal is to be clear about the strengths and limitations of our method.

• L. 260 – *How was the molar ratio of 1:1 verified as this is very difficult to do/control using unlabelled peptides. No details in the method section.*

Important point. To make it clear we have added a paragraph to the method section, in which we described how we got the 1:1 sample (Line 571 to 584). Briefly, following proteolytic digestion equal aliquots were subjected to deglycosylation. One aliquot was deglycosylated with sequential Endo H and PNGase F treatments. Another one was 'mock digested' with buffer only (no EndoH), followed by PNGase F treatment. Aliquots of these reaction mixtures were then analyzed directly, or mixed 1:1 by volume to achieve the 1:1 ratio. We rely on the fact that the samples started at equal molar ratios and volumes of each sample are matched during the short enzyme treatments, and the 1:1 mixture is created before any subsequent sample processing steps.

• L.310 ff: *As this workflow would only detect changes from oligomannose to complex or the other way around, but not changes to e.g. smaller oligomannose type structures or smaller complex structures – it can't really be concluded that there is no significant effect on processing neighbouring glycans*
This is an interesting comment. We agree that if we see all oligomannose or all complex before the mutation, we may not see differences in processing that only alter the glycan processing within those categories. On the other hand, for the majority of sites with both complex and oligomannose chains to start with, the ratio of oligomannose to complex would be a sensitive measure of changes in glycosylation, since the switch from high mannose to complex reflects the degree of processing. But we accept the point made by the reviewer, and have modified the statement accordingly. (Line 330 to 334).

• L1. 332 ff *Are all of these trimers of the SOSIP design? Why are not all of them His tagged for consistency (B41 is not)? It should be commented on how they were produced and purified.* Judging from the Method section, all of them were purified differently indicating that the experimental design has not controlled all variables.

All constructs are indeed SOSIP stabilized trimers, bearing the standard SOS and IP mutations published for BG505 by Sanders et al., Plos Path 2013. As we explain in the method section, only BG505 SOSIP is His-tagged, 327C and B41 are Avi-tagged and all other constructs were untagged. We made edits in the method section to highlight the purification methods used (Line 493 to 497, 501 to 507). To exclude the possibility of differences in glycosylation arising from differences in the purification method, we did an exhaustive comparison of Nickel and antibody affinity based purification strategies, which showed no significant differences between the purification methods (Fig 4a and 4b).

Discussion

• L1. 392: *the here discussed negative aspects of glycopeptide quantitation is not balanced as there are also a number of papers arguing that the actually glycan present on a glycopeptide does not impact its ionization efficiency e.g. Wada 2007 Glycobiology*

We agree that there is not yet consensus in the field regarding ionization efficiency of glycopeptides with same peptide backbone but different glycoforms. We have accordingly

rephrased these sentences to provide a more balanced perspective on the status of this issue (Line 432).

• LL 399 Panico et al., did not construct a glycan library per se. They just also looked at the glycomics of the sample without per se using this data as a library. Hence “several workflows” is a bit exaggerated.

We deleted this sentence since this was a workflow used by the Crispin group, and the next paragraph focuses on the data from this group.

Figures/Tables:

Figure 1C:

It is not entirely clear, what exactly each dot represents – replicates/same site on different peptides – both of it?

Each dot represents the pooled data from a single peptide. There are multiple peptides for each glycosite. So for each peptide the sum of the three glycosylation sites totals to 100%. Dots displayed at each glycosite represent same glycosite on different peptides and replicates for the same peptide in separate MS runs. Because of potential variation from run to run, we didn't combine the ion intensity peak area of the peptides across different MS raw files (Lines 651 to 654).

SI Figure 1: Why are sites 99 and 156 represented by 7 dots (replicates?) vs just one dot for the third site (176)

These three glycosites are detected on different peptides, and we could detect all three glycosites on fetuin in all MS runs. Each dot represents data from a separate run. For sites 99 and 156 the data met the threshold of 5E8 in 7 sets of data, but for the third site (176) only one set of data met this threshold. Thus only one dot was displayed at the third site (N176).

SI Figure 5: Construct names are misleading, if all of them are of the SOSIP design, add SOSIP to all names. E.g. B41 SOSIP.664

We have added SOSIP.664 to all SOSIP constructs.

Conclusion

The manuscript presents interesting and useful information about Env glycosylation, particularly the occupancy, which has only been investigated by Go et al on other Env variants. The workflow uses established techniques and the paper should probably be judged on the biological insights presented. These include occupancy measurements of the glycan sites and comparisons with strains that go well beyond Behrens. The paper will therefore be of considerable interest to those working in that area.

Thank you for the positive comments. It is also important to note that Go et. al. reported occupancy very qualitatively. For most sites, glycosylation is indicated as 0 or 1, partially glycosylated. This conclusion is consistent with 50% unoccupied, but in contrast we show that the Env is glycosylated >95% across all sites.

Reviewer #3 (Remarks to the Author):

Cao et al. described a novel method for characterization of protein glycosylation with precise and quantitative mapping of site occupancy, and demonstrated its utility on HIV-1 envelope glycoprotein, the key protein contributing to virulence of HIV and the main target of neutralization antibodies for HIV management. The method cleverly combines a series of existing methods, including O18 labeling,

endoglycosidases of endo H and PNGase F, and different proteases, in defined order to quantitatively characterize the site-specific occupancy for three forms of N-linked glycosylation: complex, high mannose (hybrid), and no glycans. The information is critical for functional studies of these glycans and their carrier proteins, which are not limited to HIV envelope glycoproteins, but to other glycoproteins encoded by a large portion of the genome regardless of the species.

We thank the reviewer for these positive comments about the method.

The method is described in detail and thoroughly characterized particularly with the efficiency of endoglycosidases. The references are sufficient, and the writing can be improved in a few places listed below.

1) The study used three different proteolysis methods for HIV-1 Env protein and each separately processed by endo H and PNGase F and finally analyzed by LC-MS in triplicates. So 9 samples are characterized by LC-MS from each HIV-1 Env protein and with 3 technical replicates, which is a careful analysis design. The biological replicates were achieved by comparing the result of regular Env trimer with those experienced different enrichment approaches and those expressed with inhibitor, which is an efficient design to validate robustness of the method and to maximize the obtained information at the same time.

There is no comment on the reason of choosing the specific combination of three different proteolysis methods. It would be helpful for future users to know whether the use of chymotrypsin, trypsin and chymotrypsin, and ‘triple digestion’ is a must or the parallel proteolysis can be simplified to two instead of three routes.

Thank you for the comments on the design, and pointing out that we did not comment on the importance of the protease conditions. They are in fact critical. Before introduction of the ‘triple digestion’, we were getting only 60% sequence coverage and detecting a maximum of 20 of the 28 glycosites, some with only a few spectral hits. So the triple digestion is clearly important, but for many less complex glycoproteins, not all three conditions would be needed. We have added comments on how to choose the specific proteolytic method for a specific glycoprotein (Line 399 to 403). Briefly, triple digestion is enough for most standard glycoproteins with a few N-glycosites, such as fetuin and invertase. For those that are heavily glycosylated, such as HIV-1 Env and some flu HA, a combination of triple digestion and chymotrypsin should be enough for detecting all N-glycosites in most cases. If not, the additional combination of trypsin and chymotrypsin is worth trying in those cases.

2) The study employed different combinations of proteases for mapping the glycosylation states. On page 7, “peptide sets” is used to describe the results of three types of possible glycosylation, i.e. high mannose, complex, and no-glycans. But it is unclear to me what protease results they represent. Is it possible to have multiple peptide sets at one particular glycosylation site in a single proteolysis scheme? If so, is this due to miss cleavage?

Yes, it is very possible to have multiple peptide sets at one particular glycosylation site in a single proteolytic digestion method, which is due to the non-specific protease enzymes, including elastase, subtilisin, and chymotrypsin, that we used in this study. In one extreme example, for site 234 there were 237 peptide sets identified, but only 33 of these met the threshold of 5E8. In fact, it is an advantage of the non-specific proteases that a ladder of peptide is produced around a glycosylation site to provide confidence in the result. If a single trypsin digestion was used site localization would effectively be based on “one hit wonders”.

3) The use of different protease in combination is one novelty for mapping the glycosylation occupancy. The observed biases in some “peptide sets” mentioned on page 7, lines 187 and 188 can be included

in supplementary with a breakdown on the used proteases. A further explanation of the apparent biases would be helpful. Will it be possible that certain proteases have preferences to cleave peptides with certain glycan structure, or their presence is stochastic?

This is a great suggestion. We have added a figure with a breakdown on the proteases in supplementary material (Supplementary Fig. 4), and a further explanation of the apparent biases (Line 209 to 210). We found that the bias observed is stochastic rather than based on the preferences of the proteases to cleave peptides with certain glycan structures. Indeed, all proteases used, including trypsin, elastase, subtilisin, and chymotrypsin generated data with some bias.

4) The method used the selectivity of endo H and PNGase F to distinguish the types of N-linked glycans, in which the high mannose structure recognized by endo H does not exclude the hybrid glycans that include both high mannose and complex glycans. This concern was discussed on page 7, line 178-181 for HIV-1 Env. It is also helpful to mention this confounding effect in the introduction and in the discussion when the method was recommended to the characterization of general glycoproteins.

As summarized in part in replies to comments of reviewers 1 and 2, we have added comments in the introduction, results and discussion to make clear that hybrid are included in the high mannose category (e.g. Line 103-106, 126, and 395).

In summary, it is a solid study and addresses an important area that concerned by a number of fields including but not limited to virology, immunotherapy, glycobiology, and signaling. The study is suitable to publish in Nature Communication with additional edition.

Again, we thank the referee for the positive comments.

Reviewers' Comments:

Reviewer #1 (Remarks to the Author):

The authors detailed response and modifications to this reviewer's comments are satisfactory for publication. Concern remains about quantification of peptides with +3 or +203, particularly given deviation from a 1:1 mixture of the two species in Figure 3. This variability adds a degree of error that is not represented in subsequent error bars. Otherwise, the method is a useful approach to distinguish complex and hybrid glycan occupancy and may be applied generally, as demonstrated on both the title protein and several others in the SI.

Reviewer #2 (Remarks to the Author):

The authors responses are appropriate and I support publication with the attention to a few minor points.

Reference error - Panico study reference 77 and 78 are the same.

Behrens has now published an article on the impact of trimerization and cleavage (<http://jvi.asm.org/content/early/2016/10/27/JVI.01894-16.abstract>) which contains info on some of their previous missing sites. Do update relevant sections and comparisons.

Also by the same group, a total mass assessment has been made of gp120 with uniform glycans suggesting an occupancy much higher than measured in the present work (<https://www.ncbi.nlm.nih.gov/pubmed/27984693>). Probably suggests an impact of different purification methods of the monomeric material but should be commented upon.

Reviewer #3 (Remarks to the Author):

The authors have addressed my concerns. A few suggestions are listed below:

1. Regarding to quantitation, the design is very thoughtful with normalization of ion injection time to even out co-eluting sample complexity as the comparing glycopeptides will migrate differently when carrying different glycosylations. The 1:1 mixing experiment demonstrated that there were some variations. One way to approve the changes observed in the samples of interest is to take the extreme case as the authors discussed; another way to approve is to use an orthogonal quantification approach such as spectra counting, which does not require any

additional experiments but just re-analyze the results. With drastic quantitative changes, any methods should pick them up.

2. Fig. S4 is very helpful for indicating the necessity of using the triple digestion. It would be better to highlight the sites would otherwise provide incorrect results in the figure.

3. Fig. S1 keys may be wrong. "No glycan" is purple in the keys but explained as grey in the legend.

4. Fig. 3a the high mannose structure treated with PNGase F alone should remain as green line, isn't it? I am not sure why it changed to purple at the end.

Responses to Reviewer comments:

Reviewer #1:

-The authors detailed response and modifications to this reviewer's comments are satisfactory for publication.

We thank the reviewer for this positive comment.

-Concern remains about quantification of peptides with +3 or +203, particularly given deviation from a 1:1 mixture of the two species in Figure 3. This variability adds a degree of error that is not represented in subsequent error bars. Otherwise, the method is a useful approach to distinguish complex and hybrid glycan occupancy and may be applied generally, as demonstrated on both the title protein and several others in the SI.

This concern was also raised by reviewer 3, who suggested that we compare our method of using peak area, with or without site specific correction for site specific detection of complex vs high mannose glycans, with the more commonly used method of using spectral counts as a method of quantitation. Indeed there are deviations of the two species in the 1:1 mixing experiment. We had reported that site-specific corrections for the results obtained for the BG505 SOSIP.664 trimer shown in Fig. 1d and Supplementary Fig 6a. were both quantitatively and qualitatively similar before and after correction (Supplementary Fig 6a and 6b). Indeed, the corrected values did not show statistical difference from uncorrected values at 27 of the 28 glycosites in BG505 SOSIP.664 trimer. The exception was site N637, which showed a statistically significant difference in the proportion of high mannose and complex type glycans, but qualitatively was no different. At the suggestion of Reviewer 3 we also evaluated the same data using spectra counts as the main quantitative method (Supplementary Fig 6c). Here we found a higher level of complex type glycans in 14 of 28 sites. We believe this bias is a consequence of the fact that the sensitivity of the Orbitrap Fusion used in this study biases against low abundance vs high abundance peptides when both yield a single spectral count (1:1) in a given spectra. We discussed this in results (Line 286-299).

Reviewer #2:

-The authors responses are appropriate and I support publication with the attention to a few minor points.

We thank the referee for the support.

-Reference error - Panico study reference 77 and 78 are the same.

Thank you, this has been fixed.

-Behrens has now published an article on the impact of trimerization and cleavage (<http://jvi.asm.org/content/early/2016/10/27/JVI.01894-16.abstract>) which contains info on some of their previous missing sites. Do update relevant sections and comparisons.

We have added the new article and updated comments in the text (See Line 84 and 441).

-Also by the same group, a total mass assessment has been made of gp120 with uniform glycans suggesting an occupancy much higher than measured in the present work (<https://www.ncbi.nlm.nih.gov/pubmed/27984693>). Probably suggests an impact of different purification methods of the monomeric material but should be commented upon.

Although the monomer is not the main focus of this study, the overall site occupancy of the BG505 gp120 monomer in the current study is approximately 86%, which we believe is consistent with the results of the study cited above. Comments on this have been added (Line 355-356).

Reviewer #3:

-The authors have addressed my concerns. A few suggestions are listed below:

We thank the reviewer for the positive comment.

1. Regarding to quantitation, the design is very thoughtful with normalization of ion injection time to even out co-eluting sample complexity as the comparing glycopeptides will migrate differently when carrying different glycosylations. The 1:1 mixing experiment demonstrated that there were some variations. One way to approve the changes observed in the samples of interest is to take the extreme case as the authors discussed; another way to approve is to use an orthogonal quantification approach such as spectra counting, which does not require any additional experiments but just re-analyze the results. With drastic quantitative changes, any methods should pick them up.

As also raised by reviewer 1, this is an important point. We had in fact initially evaluated spectral counts as an alternative method of quantitation. But because of the sensitivity of the instrument used (Fusion Orbitrap), quantitation by spectral count inherently over represents low abundance species since low intensity and high intensity spectral hits are counted the same. As suggested by reviewer 3, we have now compared the data from BG505 SOSIP.664 trimer analyzed using peak area, corrected peak area, and spectra counts, in Supplementary Fig. 6. In contrast to very minor differences that result in proportions of unoccupied, high mannose and complex type glycans using site specific correction of peak area, spectral count analysis significantly increases the apparent proportion of complex chains in half the sites, including those that are predominately high mannose as reported by others Behrens, Cell Reports, 2016, 2695 & Guttman, Structure, 2014, 974.). We thank the reviewers for this suggestion because it illustrates the importance of using peak area, and by contrast shows that the site-specific correction of complex vs high mannose proportions obtained by peak area is not needed. We also added comments to manuscript on this (Line 286-299).

2. Fig. S4 is very helpful for indicating the necessity of using the triple digestion. It would be better to highlight the sites would otherwise provide incorrect results in the figure.

Thank you. We have highlighted the sites would otherwise provide incorrect results in Supplementary Fig. 4.

3. Fig. S1 keys may be wrong. "No glycan" is purple in the keys but explained as grey in the legend.

Thank you for catching this error. We have changed it to grey in the keys of Fig. S1.

4. Fig. 3a the high mannose structure treated with PNGase F alone should remain as green line, isn't it? I am not sure why it changed to purple at the end.

Again, we appreciate the careful review and we have changed it to green line in Fig. 3a.